# Molecular control via dynamic bonding enables material responsiveness in additively manufactured metallo-polyelectrolytes

Seola Lee [1,4] ✉, Pierre J. Walker [2,4], Seneca J. Velling [1], Amylynn Chen[1], Zane W. Taylor [1], Cyrus J.B.M Fiori [2], Vatsa Gandhi [1,3], Zhen-Gang Wang [2] & Julia R. Greer [1,3]

Metallo-polyelectrolytes are versatile materials for applications like filtration, biomedical devices, and sensors, due to their metal-organic synergy. Their dynamic and reversible electrostatic interactions offer high ionic conductivity, self-healing, and tunable mechanical properties. However, the knowledge gap between molecular-level dynamic bonds and continuum-level material properties persists, largely due to limited fabrication methods and a lack of theoretical design frameworks. To address this critical gap, we present a framework, combining theoretical and experimental insights, highlighting the interplay of molecular parameters in governing material properties. Using stereolithography-based additive manufacturing, we produce durable metallo-polyelectrolytes gels with tunable mechanical properties based on metal ion valency and polymer charge sparsity. Our approach unveils mechanistic insights into how these interactions propagate to macroscale properties, where higher valency ions yield stiffer, tougher materials, and lower charge sparsity alters material phase behavior. This work enhances understanding of metallo-polyelectrolytes behavior, providing a foundation for designing advanced functional materials.

Metallo-polyelectrolyte Complexes (MPEC) are a class of soft materials that exhibit unique mechanical and physical properties through reversible electrostatic interactions between dynamic crosslinkers (multivalent metal ions) and charged polymer chains. MPECs consist of polyanions that are electrostatically crosslinked by labile metal cations with secondary metal–ligand coordination[1–3]. In concert with the polymer chain conformations that arise from electrostatic interactions, these molecular-level processes govern global phenomena in the gels. For example, the local coordination environment yields distinct stiffness response[4,5], while the oxidation state of the metal ion has a significant impact on the ionic conductivity[6]. This molecular-level control offers a wide range of opportunities to explore for a range of applications—from filtration to biomedical devices and sensors[7–10].

The multiscale physical behavior of MPEC gels, ranging from local coordination environment at the atomic level to mesoscale polymer conformation to macroscopic material properties, is strongly affected by the chemical nature of metal complexation[1,11,12], solution pH[13] and solvent quality[14]. The knowledge gap between the molecular-level chemistry of the dynamic bonds and the continuum-level material properties, combined with the lack of mechanistic insight from theory and simulations, have limited the development and utilization of this class of materials in real-world applications[2]. Existing state-of-the-art fabrication methods are

[1]Division of Engineering and Applied Science, California Institute of Technology, 1200 California Boulevard, Pasadena 91125 CA, USA. [2]Division of Chemistry and Chemical Engineering, California Institute of Technology, 1200 California Boulevard, Pasadena 91125 CA, USA. [3]Kavli Nanoscience Institute, California Institute of Technology, 1200 California Boulevard, Pasadena 91125 CA, USA. [4]These authors contributed equally: Seola Lee, Pierre J. Walker. ✉ e-mail: seolalee@caltech.edu

typically complicated and require multiple steps of solution-based metallo-polyelectrolyte synthesis[2,15] to produce samples, most of which have been reported to lack long-term stability, suffering from inhomogeneity, and in some cases requiring the addition of non-dynamic covalent crosslinkers for synthesis[11,16]. The use of theoretical methods to gain intuition of and alleviate these experimental complications is also hampered by computational challenges, arising from the large range of length and time scales involved[2,12,17,18].

To close this substantial knowledge gap between the molecular-level chemistry of dynamic bonds and the continuum-level material properties, we fabricated homogeneous, stable, and long-lived poly(-acrylic acid) (PAA) MPEC gels using a single-step stereolithography and conducted a theory-guided, physically-informed multi-scale study of MPEC gels. Through holistic material characterization, we probe the relevant length scales to corroborate theoretical and computational predictions. Based on these findings, we present a roadmap that outlines the effect of chemical composition on the MPEC gel properties that can be used to tailor their functionality and responsiveness at the material level. The overall framework and chemical species explored in this work are common in soft materials formed with dynamic bonds, which renders our findings readily applicable to different material systems.

## Results

### Chemically-tunable synthesis of MPECs

We synthesized acrylate-based MPECs using a facile, single-step fabrication method via Liquid Crystal Display (LCD)-stereolithography. We first prepared a homogeneous photoresin solution that consisted of acrylic acid (AA) monomers and sodium acrylate (SA) co-monomer buffer that serve as chain builders, combined with metal ion species as dynamic crosslinkers. As illustrated in Fig. 1a, PAA was formed by photopolymerization, initiated by addition of ethyl (2,4,6-trimethylbenzoyl) phenylphosphinate (TPO-L) photoinitiator and tartrazine UV blocker (yellow-dye). To produce environmentally-stable, high-longevity materials, glycerol and water were used as co-solvents to suppress gel dehydration. For fair comparison, most of synthesis parameters, except the metal valency and system pH, were kept consistent between the gels. The total monomer concentration was kept constant at 8.8 M to ensure consistent polymerization. Concurrently, the maximum number of potential carboxylate–metal cation bonds was conserved at 2 mol% of monomer concentration, irrespective of the number of available binding sites on the polymer chains (see details in Methods and Supplementary Tables 1 and 2). The 3D-printed gels, after post processing, maintained their mass at 85% − 90% of the as-printed state for more than 90 days (Supplementary Fig. 2). Themogravimentric analysis (TGA) confirms that all equilibrated gels,

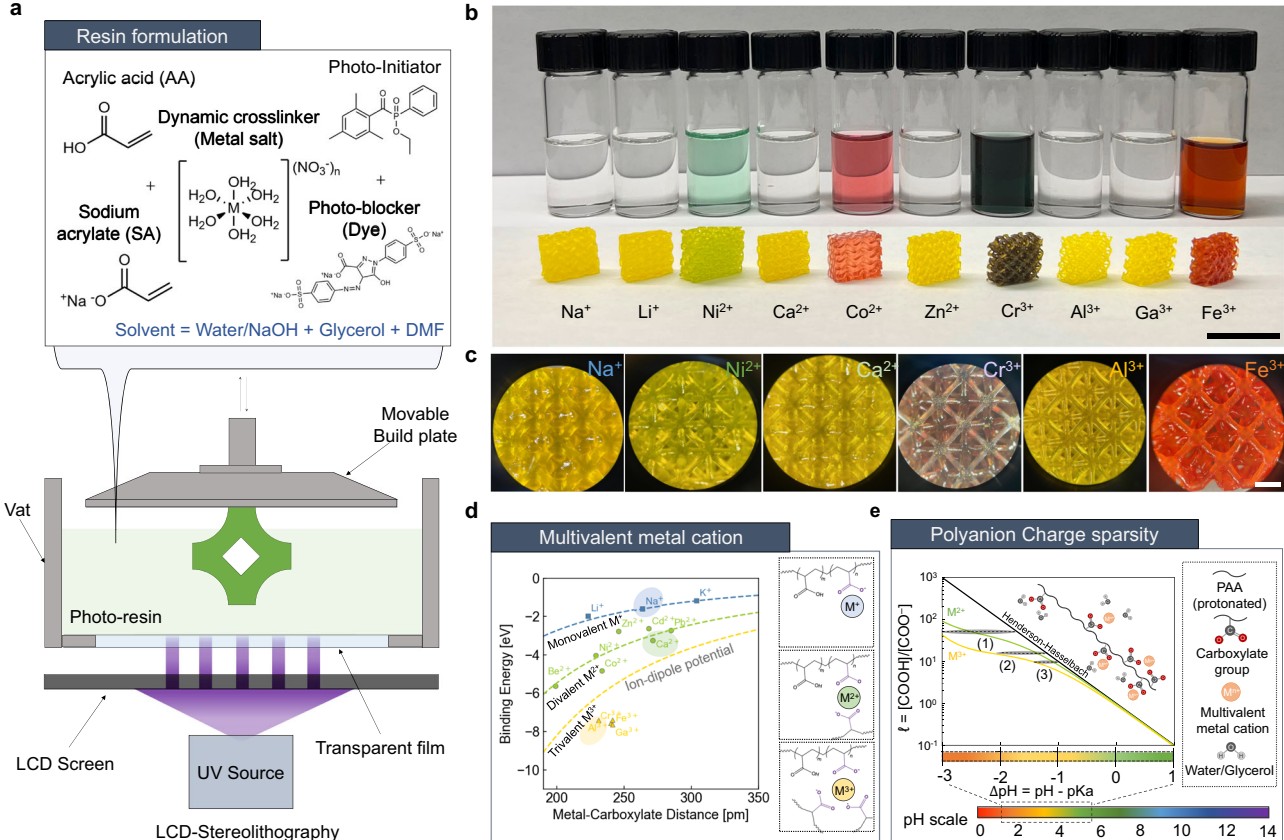

**Fig. 1 | Fabrication of Metallo-polyelectrolytes with different metal ions.**
**a** Single-pot synthesis of MPEC using stereolithography. **b** Optical images of synthesized photo resins (without tartrazine to show coordination color) and 3D-printed lattices (with tartrazine) with different metal ions. Scale bar 15 mm.
**c** Magnified images of 3D-printed octet lattices, showing multiple unit cells. Scale bar 2 mm. **d** Binding energy vs. metal-carboxylate separation of multivalent metal ions printable with the developed fabrication method. Dashed curves represent theoretical metal cation-PAA dipole interactions. The inset highlights how the different charges of metal cations lead to distinct coordination numbers around the metal center. **e** pH versus charge sparsity of polyanion chains, with varying cation valency of fixed concentration (2 mol%). Strong association of multivalent metal cations pushes the equilibrium forwards below the Henderson–Hasselbach theory in the range of pH of interest.

irrespective of the metal ions and the system pH chosen, contain ~14% ± 2.14% water by weight. This uniformity underscores the consistency in solvent content across all samples (Supplementary Discussions 6). X-ray fluorescence mapping confirmed that metal crosslinkers were homogeneously distributed throughout the sample. This methodology is generic and can be used to produce a broad range of material with different compositions through selection of an aqueous anionic monomer and various metal salts in the initial resin. Figure 1b demonstrates several multivalent metal species used to synthesize MPECs into 3D structures with $\mu$m to mm resolution and complex geometries.

To screen for candidate metal nitrate salts we used quantum Density Functional Theory (DFT) to calculate the binding energy between various metal cations and carboxylate (acetate) (see Methods and Fig. 1d). The binding energy scales linearly with the cation valency, highlighting control over both the coordination number of complexed polyanion sites[19–21] and the binding energy. We observe that metal ions with available d-orbitals, such as $Ni^{2+}$ and $Cr^{3+}$, typically deviate from the expected scaling behavior due to the formation of overlapping $\pi$-orbitals (see Supplementary Discussions 1). We focus on three representative hard-ions for each valency, $Na^+$, $Ca^{2+}$, and $Al^{3+}$, to isolate the effects of valency without the concomitant effects of d-orbital interactions. Discussions relating to the metal identity can be found in Supplementary Discussion 2. Owing to the low binding energy of $Na^+$ relative to protonation, pure PAA gels and $Na^+$-MPEC gels are used as a system control, depending on the pH range explored. The weak electrostatic interactions of $Na^+$ ions allow the monovalent gel to serve as a reference system where entanglement is the major contributor to the material response. For comparison, non-dynamic PAA-gels covalently crosslinked with N,N'-Methylenebisacrylamide (MBAA) were also fabricated to demonstrate the contribution of the dynamic bonds. For the remainder of this article, each crosslinker is represented by a consistent color except where otherwise specified: blue for $Na^+$, green for $Ca^{2+}$, yellow for $Al^{3+}$, and gray for the MBAA.

Another key parameter is the pH of the resin, which was controlled using nitric acid and sodium hydroxide, complementary to the sodium acrylate buffer and nitrate salts. During printing, each ~5 um-thick layer is saturated in the resin with exposure time of 30 s, allowing for equilibration with the solution. To minimize pH-dependent chain conformation effects induced from anion repulsion and the changes in hydrogen bonding[22], we maintained the pH in our experiments below the pKa of polyacrylic acid (~4.5). Fourier Transform Infra-red Spectroscopy (FTIR) confirmed that the change of hydrogen bonding was minimal within the range of pH explored (Supplementary Discussions 4). The high ionic strength of the gel[23,24], at pH far from the pKa of the system, allows us to correlate the pH with charge sparsity of polymerized polyanion following the Henderson–Hasselbach (HH) equation, $\langle \ell \rangle$ = [COOH]:[COO$^-$]. The strong association of multivalent metal salts with the charged sites on the polymers shifts the deprotonation equilibrium forwards[25,26], reducing the charge sparsity of the polyanion relative to the HH expectation. We account for this effect using a modified HH equation, as shown in Fig. 1e (see Supplementary Discussions 3 for derivation). Under the experimental conditions, the modified HH equation demonstrates the availability of sufficient number of binding sites on the polymer backbone to complex with the available metal ions ([COO$^-$] $\gg n$[M$^{n+}$]). Figure 1e conveys the three pH regimes experimentally explored in this work: (1) low (1.5 < pH < 2.5), (2) intermediate (2.5 < pH < 3.0), and (3) high (3.0 < pH < 3.5). Using pH measurements of the photoresin as a proxy, these values correspond to the range of polyanion charge sparsity $\langle \ell \rangle$, of ~100:1 –10:1. (see Supplementary Table 3).

The presented fabrication platform lends itself to modulating the two key levers, metal valency and pH, to understand the extent of molecular control on the material properties. Another lever, albeit less impactful and more challenging to control, is the solvent content, which, for the sake of conciseness, is examined in greater detail in Supplementary Discussions 7.

## Structural bonding of metal–carboxylates
In Fig. 2a, representative FTIR spectra are shown for MPEC samples with $Na^+$, $Ca^{2+}$, and $Al^{3+}$. These spectra indicate that metal ions associate with the polymer, forming stoichiometrically charge balanced complexes. The polyacid nature of MPEC gels gives spectral character for carboxyl (R-COOH) and carboxylate (R-COO$^-$) functional groups, with the local modes of R-COOH/COO$^-$ groups identified in Fig. 2a, at characteristic carboxyl absorption at ~1760 cm$^{-1}$ (monomer) / ~1700 cm$^{-1}$ (dimer) and alcohol C-OH stretch at ~1240 cm$^{-1}$. The carboxylate anion exhibits clear bond-and-a-half character with asymmetric and symmetric stretches at ~1545 cm$^{-1}$ and ~1410 cm$^{-1}$, respectively. Pure PAA gels exhibited the expected carboxyl stretches only, which indicates that the polymer is completely protonated (see Supplementary Discussions 4). The presence of symmetric and asymmetric modes of the carboxylate anion at pH $\lesssim$ 2.5 indicates complexation of metal cations in MPECs. The structural mode of complexation of each metal species was revealed by the separation of the R-COO$^-$ stretches, using sodium polyacrylate salt at pH = 13 as a spectral reference[27,28]. Taken together with the hard ionic nature of the selected metal cations, these results support a bidentate chelation of each metal cation, with direct coordination of the metal species by the polymer. Maintaining charge neutrality requires correspondence between the number of coordinating carboxylates and the metal cation valency, respectively. Our interpretation of the local coordination environment is further supported by DFT simulations of pure acetate anions (CH3COO$^-$) bonding to each dynamic crosslinker and their associated IR signature, irrespective of the presence of solvent (see Supplementary Discussions 4).

## Thermal characterization
Differential thermal analysis was conducted at a low-ramp rate of $\sim 5\frac{°C}{min}$ from −30 °C to 200 °C to identify phases and phase transitions of the gels at ambient conditions. The Differential Scanning Calorimetry (DSC) thermogram (Fig. 2b, bottom) reveals the existence of a water solvation shell that participates in two processes: solvating the polymer and contributing to the metallo–polyelectrolyte complexation. The MPEC gels show a distinct and repeatable non-crystallization exotherm at 140 °C < $T$ < 170 °C prior to an evaporative endotherm, caused by desolvation and evaporation of water in a manner consistent with the literature for the dehydration of related oxalate compounds within a similar temperature range. This evaporative thermal signature, with associated nucleation of vapor pockets, persists even when other co-solvents are removed. It is observable in dimethylformamide free MPECs synthesized by use of a water soluble photoinitiator, lithium phenyl (2,4,6-trimethylbenzoyl) phosphinate (Supplementary Discussions 7.1.2). Optical micrographs in the insets show the presence of water vapor pockets within the gels, with no other observable phase transitions present. Thermogravimetric analysis of MPECs confirms that the extent of loosely bound (≈15%) and solvated (≈5%) water remains in all equilibrated gels, which allows for segmental polymer motion and ion exchange (Supplementary Discussions 6).

Dynamic Mechanical Analysis (DMA) demonstrates visually no dependence of the thermal-phase transitions on metal valency: glassy regime below 0 °C followed by $10^2$–$10^3$ fold decrease in the storage modulus ($E'$) around 5–50 °C, indicative of the glassy-to-rubbery transition (Fig. 2b, top). The rubbery phase initiates above 50–70 °C, characterized by enhanced polymer chain mobility at higher temperatures. The thermal signatures from DSC and DMA indicate the absence of gel melting until the onset of water evaporation at 170 °C. Water evaporation embrittles the material, increasing $E'$ by a factor of 20–30 and leading to unstable measurements of the loss factor (tan($\delta$)) at temperatures higher than 170 °C (Fig. 2b, c). This signifies

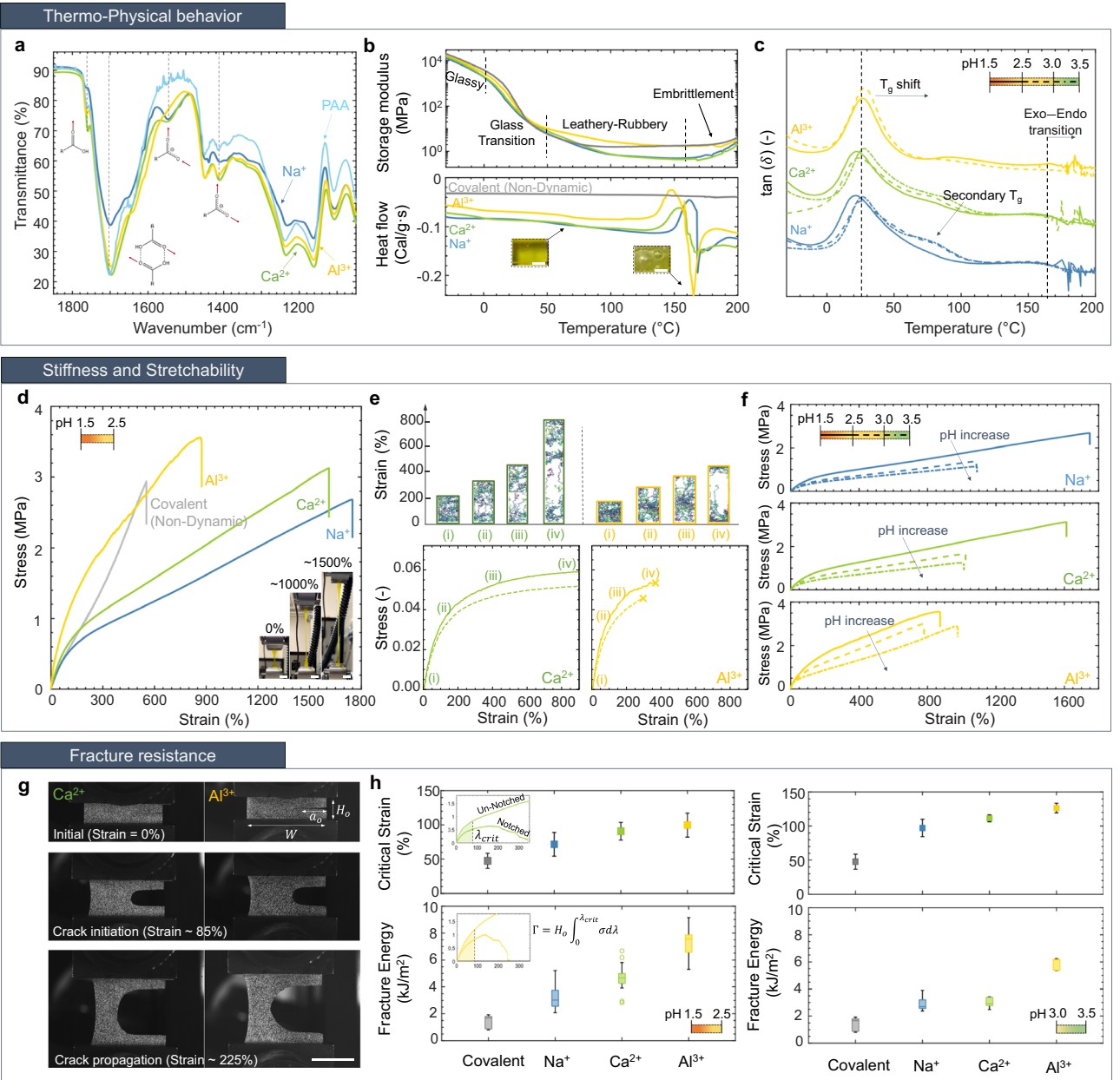

**Fig. 2 | Material-level properties of the additively manufactured MPECs cross-linked with Na⁺, Ca²⁺, and Al³⁺ in all plots. a** FTIR spectra exhibit signatures of both carboxyl and carboxylate functional groups, with the later being associated with bidentate chelation of Na⁺, Ca²⁺, and Al³⁺. **b** Storage modulus with overall DSC thermogram and **c** Loss factor of MPECs at different pH (curves are vertically shifted for visual aid). Inset: Optical images of water pocket formed within gel. Scale bar 2 mm. Different linestyles on the pH scale in inset represents pH regime of the characterized gels. **d** Experimental stress−strain data of gels crosslinked with each metal crosslinks versus with MBAA under uniaxial tensile loading. Inset: Optical images of Ca²⁺ stretchability. Scale bar 10 mm. **e** MD simulation snapshots (top) and stress−strain data (bottom) for different valency (left: divalent, right: trivalent) and charges parsity (solid: high sparsity, dashed: low sparsity) under uniaxial tensile loading. **f** Experimental stress−strain data as a function of pH, which shows degradation in mechanical performance in increasingly alkaline environments. **g** Optical snapshots obtained at strains of 0%, 85%, and 225% during quasi-static fracture experiments on MPEC samples subjected to pure shear loading. Scale bar 20 mm. **h** Box plots depicting fracture energy and critical strain for crack initiation calculated based on stress−strain data (inset) during fracture experiments for charge-sparse MPEC (left panel) and charge-dense MPEC (right panel).

the importance of the solvent phase in mediating the polymer motions in MPEC gels. Figure 2c shows that the peak of tan($\delta$) occurs at 15−25 °C for all samples, with a broad transition range in 0−50 °C, characteristic of glass transition $T_g$ in radical acrylate chemistries. In contrast to the similar phase transitions among different metal species, increasing pH from 1.5 to 3.5 induces a noticeable shift in $T_g$ and the emergence of a secondary transition at 70−100 °C, (as seen in Fig. 2c). This $T_g$ shift with a wider distribution implies a significant effect of pH on polymer morphology and uniformity.

## Mechanical characterization
We conducted uniaxial tensile experiments on dog bone-shaped specimens at a constant strain rate, $\dot{\epsilon} \approx 0.03\,s^{-1}$ at 25 °C. In Fig. 2d, we show the stress−strain data for MPEC gels infused with different metal ions within the low pH regime. This figure demonstrates a valency-dependent effect: gels with higher valency metal ions have greater stiffness and strength, and a lower stretchability. The Young's modulus, estimated as the slope of the initial linear regime up to 10% strain, increases from 0.92 MPa for Na⁺-MPEC gel to 1.30 MPa for Ca⁺-MPEC

gel to 2.82 MPa for Al$^+$-MPEC gel. The stresses and strains at rupture are 2.68 MPa and 1750% for the Na$^+$-MPEC gel, 3.12 MPa and 1600% for the Ca$^{2+}$-MPEC gel, and 3.56 MPa and 870% for the Al$^{3+}$-MPEC gel. These results reveal that despite the maximum number of potential carboxylate–metal bonds being kept constant for all gels, local functionality induces a noticeable difference in mechanical response. The covalently-crosslinked gel, shown in Fig. 2d, has a similar elastic modulus to the mono- and divalent gels, most probably due to similar molecular network connectivity, which governs the material properties at low strains[29]. The permanent nature of covalent bonds leads to stiffening of the gel at larger strains and results in limited stretchability of 550%. This implies that the dynamic ion crosslinkers can dissociate at the time scale of the applied deformation, allowing stretched chain relaxation and a local slip-and-stick motion of bonds under stress (Supplementary Fig. 4).

In Fig. 2e (solid lines), the mechanical response of the experimentally-equivalent polyelectrolyte gel networks in the low pH regime obtained from coarse-grained Molecular-Dynamics (MD) simulations is shown. These simulations qualitatively reproduce the trends of metal ion valency on gel stiffness. The evolution of the polymer network at different strains (Fig. 2e, top) reveals that polymer chains in the divalent gels are able to re-orient along the loading direction during uniaxial deformation; in the trivalent gels, the polymer chains are more-closely packed and tend to form large percolating structures through the dynamic crosslinks (clusters) that inhibit re-orientation. This behavior is supported by the observation that trivalent gels maintain a higher fraction of inter-crosslinked metal ions, $f_{inter}$ than that of divalent gels under high strains (Supplementary Fig. 5).

The MD simulations predict that reducing the charge sparsity on the polyanion (increasing pH) leads to a reduction in $f_{inter}$ relative to the high sparsity system, with a corresponding reduction in gel stiffness (dashed lines in Fig. 2e, bottom). This is consistent with experimental results, shown in Fig. 2f. Gels with charge-dense polyanions exhibit a 2× reduction in modulus and tensile strength and a concomitant decrease of tensile strain from >1600% to 1000% for Na$^+$- and Ca$^{2+}$-MPEC samples. The stiffness and stretchability of Al$^{3+}$-MPEC gels are reduced by a lesser amount.

To probe the effects of the molecular-level controls on the polymer network topology, we conducted pure shear fracture experiments of MPECs. Using a notched thin gel sheets in MTS load frame (MTS Systems Co., Eden Prairie, MN), we followed the methodology first developed by Thomas, Rivlin, and Lake[30,31] and more recently adapted to probe fracture of tough hydrogels[32] (Details in Methods and Supplementary Fig. 6). During the experiment, a thin MPEC sample with dimensions ($W \times H_0 \times th$) of $40 \times 10 \times 1.5$ mm$^3$ and an initial notch ($a_0$) of 16 mm ($a_0/W = 0.4$) was stretched uniaxially at a strain rate of $\dot{\epsilon} \approx 0.02$ s$^{-1}$, to induce crack propagation from the notch until complete sample rupture (Fig. 2g). We used Digital Image Correlation (DIC) to capture the exact event of crack initiation, defined as critical strain ($\varepsilon_c$), which allowed for calculating the fracture energy, $\Gamma = H_0 \int_0^{\varepsilon_c} \sigma d\lambda$.

In Fig. 2h, we show the $\Gamma$ and $\varepsilon_c$ for MPEC samples crosslinked with each metal ion and demonstrate that the fracture energy increases with stiffness. In a range of pH of 1.5–2.5 (left panel), the Al$^{3+}$-MPEC gels achieve fracture energies of $7.32 \pm 1.04$ kJ/m$^2$, ≈50% greater than that of the Ca$^{2+}$ MPEC, whose fracture energy is $4.66 \pm 0.98$ kJ/m$^2$ and ≈120% higher than that of the Na$^+$-MPEC gel, $3.27 \pm 0.94$ kJ/m$^2$. We observed a similar trend for higher pH (right panel). Stiffer gels that correspond to MPECs with highest valency (Al$^{3+}$-MPEC gels) are also the toughest. Additionally, the critical strain for crack propagation correlates with fracture energy. As the functionality of the dynamic crosslinker increases, gels become stiffer and tougher, exhibiting greater resistance to crack initiation. We find all MPEC gels are 2–10 times tougher than their covalently crosslinked counterparts.

## Discussions

The experiments in this work reveal different material behavior of metal ion-coordinated polyelectrolyte gels with distinct molecular characteristics: metal ion valency and polymer charge sparsity. To gain insights into how molecular-level interactions govern properties at the material level, we investigated molecular interactions using a multi-scale approach—from the local coordination environment to gel-network topology as a function of synthesis parameters.

MD simulations provide a useful platform for probing the impact of dynamic bonds on the local coordination environment and gel network topology. Figure 3a shows the relaxation times of the different modes within the system and is consistent with the tendency of higher-valency ions to form stronger bonds, which results in longer ion-pair relaxation times, $\rho_{ion}(0, t)$ by almost an order of magnitude. Polymer relaxation times, defined here using the end-to-end vector auto-correlation function, $\rho_{poly.}(0, t)$, has a similar trend with valency. The longer bond lifetime of the higher-valency ions and polymer chains imposes a larger energy barrier for polymer motion and impedes chain relaxation. The most conspicuous manifestation of the inhibited polymer motion is the plateau modulus of the polymer in the rubbery state (Fig. 2b). Compared to the theoretical limit of the plateau modulus for purely entangled polymer chains ($G_e \simeq 0.12$ MPa), pure PAA and Na$^+$-MPEC gels at all pH have almost identical values owing to the lack of crosslinking (see Supplementary Fig. 8). Multivalent gels deviate from this limit where the plateau modulus scales with valency (Fig. 3b). As expected, covalent gels observe the most-limited polymer motion available with the largest plateau modulus.

Figure 3a, top also reveals that the ion-pair relaxation, $\rho_{ion}(0, t)$, is affected by changes in polyanion charge sparsity. This arises from the increased competition between available sites and the metal ions. However, this decreased ion-pair relaxation time is not significantly large, in contrast to the polymer end-to-end vector relaxation time, $\rho_{poly}(0, t)$, as shown in Fig. 3a, bottom. This trend is consistent with the fact that plateau modulus was not significantly affected by the pH of the gels (Fig. 3b and Supplementary Fig. 8). Reducing the charge sparsity dramatically impacts the polymer relaxation times (Fig. 3a, bottom). The relaxation times of the multivalents gels significantly decrease, approaching the relaxation times of the monovalent gels. This trend is consistent with the results shown in Fig. 2f where all stress-strain curves collapse to that of a Na$^+$-MPEC gel. This non-intuitive shift with decreasing sparsity appears to be correlated with the evolution of network topology, comprised of intrachain-molecular and interchain-molecular junctions and loops.

Following the approach of Semenov and Rubinstein[33] (see Methods and Supplementary Discussions 5 for details), we estimate theoretically the impact of charge sparsity on the fraction of interchain-crosslinks and compared these predictions (lines) to the simulated results (symbols), shown in Fig. 3c. The higher functionality of a single trivalent ion leads to higher fraction $f_{inter}$ compared to a divalent ion. We observe optimal charge sparsity to occur at a maximum fraction of interchain-crosslinking metal ions where a balance between entropic and enthalpic effects is struck. The experimental system described in this work contains an excess of charged sites relative to metal ion concentration, a regime where entropic effects dominate (left region of Fig. 3c).

The interchain-crosslinks play a key role in the formation of gel network. In Fig. 3d, we show the largest cluster size normalized by the total number of particles as a function of charge sparsity as obtained from MD. The simulations indicate that the di- and trivalent gels form clusters which span nearly all polymer chains within the system, with trivalent gels having the narrowest cluster distribution resulting from stronger binding and slower relaxation. No pathway to directly crosslink two chains exists in the monovalent gels, with interchain entanglements being the only mechanism to form clusters, leading to smaller clusters. The trend of the cluster size with respect to the charge

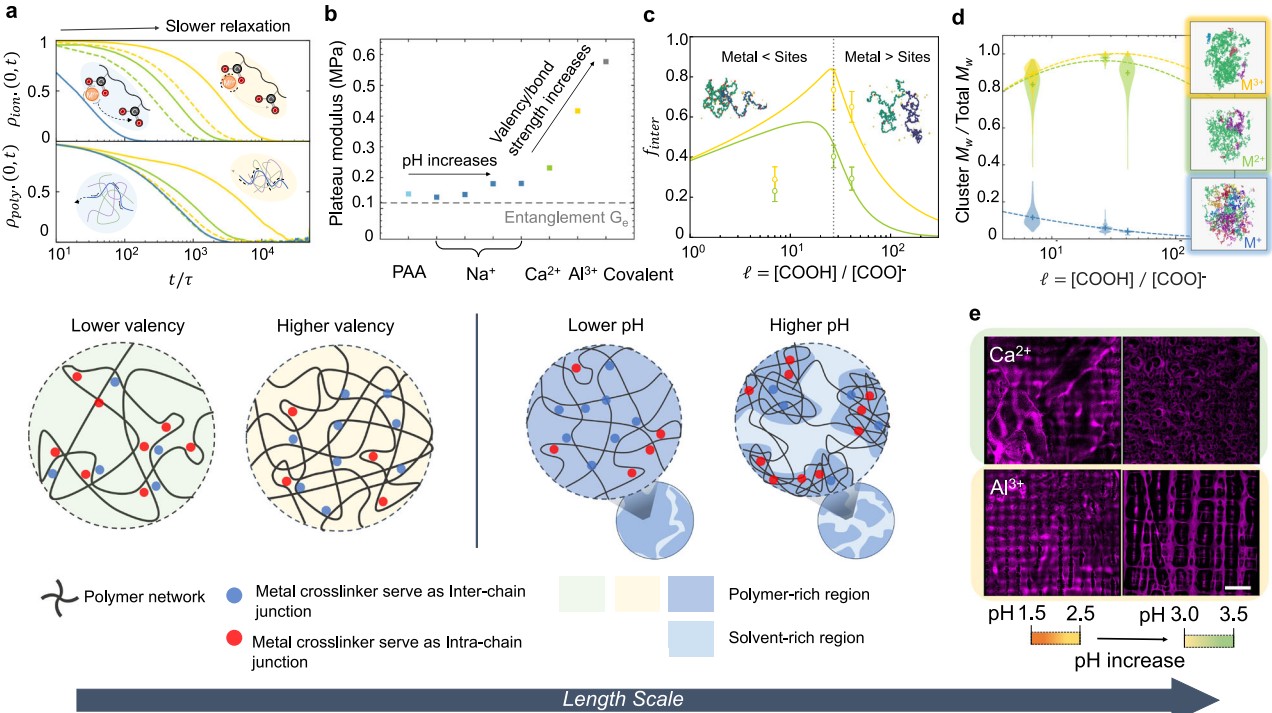

**Fig. 3 | Ionic and charge sparsity effects on gel microstructure.**
**a** Autocorrelation functions of the ion-pair (top) and polymer (bottom) end-to-end vector for MPECs with different ion valencies and charge sparsity obtained from MD simulations: solid lines correspond to the low pH and dashed ones correspond to high pH. Higher valency induces slower relaxation for both ion-pairing and polymer relaxations. **b** Effects of pH and metal ion valency on plateau modulus of MPEC in the rubbery state. Plateau modulus of MPECs were measured from Fig. 2b and the entanglement modulus was analytically estimated (see Supplementary Fig. 8). **c** Theoretically- (lines) and computationally- (markers with 95% confidence interval) predicted fraction of inter-crosslinked ions for divalent (green) and trivalent (yellow) systems at different charge sparsity. Dotted line corresponds to the sparsity where the metal ions perfectly neutralize the polyanion. **d** Violin plot of simulated cluster distribution in MPECs. Wider distributions represents larger fluctuations of the cluster size. MD snapshots illustrate representative clusters for each valency. **e** Confocal microscopy conducted at the depth of $50 \pm 25$ um on as-printed thin films of MPECs showing Poly(acrylic acid) autofluorescence (purple) at 640 nm where negative void space corresponds to fluid filled regions. Due to instantaneous process of vat-polymerization, the voxelized printing pattern ($50\,\mu m \times 50\,\mu m$) is locked in the topology for both $Ca^{2+}$ and $Al^{3+}$-MPEC. Higher pH of the gels demonstrating severe phase separation of fluid filled regions from polymeric region. Scale bar for all images 150 um. The lower-panel schematic represents the effects of metal ion valency and charge sparsity on the distribution of inter vs. intrachain-crosslinks and gel microstructure.

sparsity has a similar trend to the fraction $f_{inter}$ in Fig. 3c, suggesting that the distribution of interchain-crosslinks in the gel network represents the driving force for polymer topology evolution. The distribution of these clusters also widens with decreasing sparsity, which may contribute to the faster polymer relaxation times.

Confocal microscopy images of the additively manufactured MPEC samples with $Ca^{2+}$ and $Al^{3+}$ (Fig. 3e, up-down) captures the effect of valency and pH on the gel microstructure. In higher valency systems, the gel retains the voxelized printing pattern ($50 \times 50\,\mu m^2$), indicating the slower ion and polymer relaxation freezes the microstructure in-place as it polymerizes. The larger fraction $f_{inter}$ and stronger binding of higher valency systems result in enhanced stiffness[34] and toughness[35] due to efficient distribution of load across the chains during mechanical deformation. However, the suppressed chain and network mobility induces reduced material-level extensibilty and favors chain scission as a failure mechanism under fracture loading, instead of chain extension or crack deviation within the network[36]. The effect is apparent from the different crack morphology during fracture experiments: $Ca^{2+}$ samples exhibit significant crack blunting even at 225% strain; crack tips in $Al^{3+}$-MPEC gels become sharper immediately after crack opening Fig. 2g. With increasing pH, the polymers are able to relax faster and diffuse away from the original printing pattern, as observed visually in the right panels of Fig. 3e. This faster relaxation, coupled with a reduced fraction $f_{inter}$ leading to more-mobile clusters, is responsible for the gels weakening under global mechanical perturbation and trading elasticity for a more

viscoelastic/plastic response as verified from the stress relaxation and recovery experiments (see Supplementary Fig. 12).

With the mechanistic insights drawn from the presented experiments, simulations, and theory providing a comprehensive understanding of how molecular-level bonding mechanisms propagate to the material-level properties of MPEC gels, we have identified and demonstrated the role of metal ion valency and charge sparsity on their deformation and fracture response. To layout the full parameter space of MPECs, we have constructed phase diagrams for the gels with each metal ion valency and polyanion sparsity using an associative mean-field theory (Fig. 4a) (see Supplementary Discussions 5 for details). These phase diagrams help identify conditions where the gel exists as a single phase and where the system will split into two-phases, with the boundaries denoted by the colored contours. The insets in Fig. 4a (and in Supplementary Fig. 11), obtained from gel dissolution in water, demonstrate the typical supernatant-coacervate phase split observed in polyelectrolytes (seen in both divalent and trivalent gels). The 'notch' at low metal ion concentrations in the phase diagram of the trivalent system arises from a gel-gel phase split, where one phase is more concentrated in trivalent metal ions. This behavior is unusual and serves as a way to discover and elucidate material properties (see more experimental results in Supplementary Fig. 14 - 16).

From the developed theory, the impact of pH can be inferred from the polymer charge sparsity. A reduction in the charge sparsity of the polymer dramatically increases the area of the two-phase

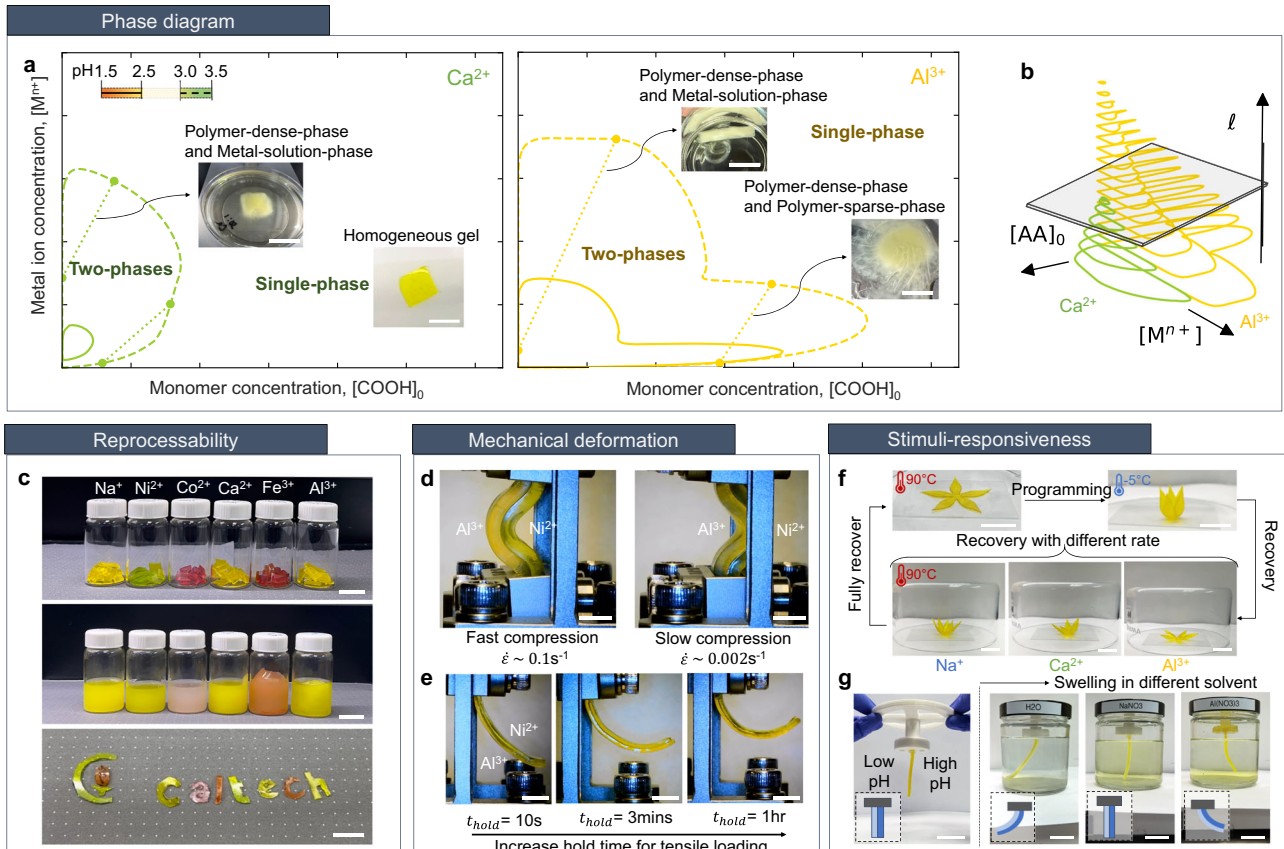

**Fig. 4 | Phase diagram and multi-functional applications. a** Phase diagram for polyelectrolytes + divalent salt (green) and trivalent salt (yellow) at different pH. Solid lines represent the phase envelope, dashed lines represent the gel curve and dotted curves represent tie-lines. Inset: Optical images of gels at each phase. Scale bar 20 mm. **b** Through-the-height contour plots of MPEC phase diagram. Increase in charge sparsity, $\ell$ (low pH) reduces the size of phase envelope. **c** Reprocessability of MPECs. Printed MPEC with various metal cations were dissolved in water and recasted into different shapes. Scale bar 10 mm. **d** Switch in buckling direction for a $Al^{3+}$-$Ni^{2+}$ bilayer beam under different strain rate. Scale bar 4 mm. **e** Different curvature of a $Al^{3+}$-$Ni^{2+}$ bilayer strip induced from mismatch in residual strain upon the release from tensile loading. Scale bar 4 mm. **f** Heating-induced shape memory behavior of a flower-shaped MPECs. Different metal ions induce different rate of recovery. Scale bar 20 mm. **g** Solvent-induced swelling of a bilayer beam, composed of $Na^+$-MPEC with low-pH (left) and high-pH (right). To preserve the gels in solvents, $Na^+$-MPEC samples were partially crosslinked with MBAA (0.01 mol%). Distinct swelling degree of each material induces different shapes of beam in various solvents. Scale bar 30 mm.

region, while retaining the overall shape. The through-the-height contour plots shown in Fig. 4b demonstrate that at sufficiently high charge sparsities (low pH), the two-phase region disappears. This is primarily a result of the decreasing electrostatic correlations. The alteration of gel microstructure with changes in pH shown in Fig. 3e reflects that, within fabrication conditions explored herein, our 3D-printed MPEC can achieve a state where a dense polymeric network (purple) and the surrounding solvent/dilute phase (black) can coexist. The observed tendency was corroborated by macroscopic changes in opacity of the printed MPEC gels (see Supplementary Fig. 9 and 10) and faster dissolution rates for gels with lower acidity (see Supplementary Fig. 11).

With an understanding of the full parameter space of MPEC gels, we find that these chemical levers offer an abundance of potential applications. As an example, the solubility of MPEC gels, demonstrated in Fig. 4a and Supplementary Fig. 11, shows the potential capability of MPECs to be reprocessed using solvent. The reversible nature of the dynamic bonds allows MPEC gels to be reprocessed when dissolved in a hydrophilic solvent or heated to an elevated temperature above $T_g$. In Fig. 4c, we demonstrate how the printed gels were reshaped into new geometries after being dissolved in water and casted in a silicone mold.

Since the molecular choice of gels significantly affects the homogeneity and rates of dissolution, the reprocessability of MPECs can be optimized with the guidance the phase diagrams on Fig. 4a; i.e. higher pH gels dissolve faster and higher metal concentrations for homogeneous dissolutions of trivalent gels.

The rich dynamic properties, enabled by the molecular controls, open new avenues to exploit dynamic bonding in flexible structures where the interplay betweeen mechanical instabilities and viscoelasticity can be used to control nonlinear deformation. Motivated by the work from Janbaz et al.[37], we exploited the different combinations of viscoelastic properties in MPEC gels and explored the instability modes of a bilayer beam under different strain-rates under compressive loading. Stress relaxation experiments revealed that $Al^{3+}$-MPEC gels have higher instantaneous modulus with slower relaxation whereas divalent MPEC gels undergo faster relaxation with higher equilibrium modulus than trivalent (see Supplementary Fig. 12). This distinct relaxation of each valency gels leads a bilayer beam, made by laterally attaching $Al^{3+}$ and $Ni^{2+}$ gels, to become highly strain-rate dependent; the construct predictably buckles to the left at fast loading and right at slow loading as shown in Fig. 4d (more details in Supplementary Discussions 8). As the buckling direction switch occurs due to

the shift in the softer-stiffer layer in the beam with respect to loading timescale, various combinations of metal ions in each building block will allow wide control over such deformation pathways. Similarly, different recovery strains of MPEC gels upon the release from tensile loading can be exploited to induce mismatch strain between the layers. The developed mismatch strain leads to different beam curvature of the strip as shown in Fig. 4e.

Finally, the responsiveness of the materials to different stimuli (i.e. heat or solvent) can be tuned depending on their molecular choices. Different rates of shape-memory behavior of MPEC gels depending on metal valency were demonstrated in Fig. 4f. Samples were deformed ('programming') at elevated temperature (90 °C) for 5 s and cooled down to lower temperature (−5 °C) to fix the programmed shape. Upon returning to 90 °C, MPEC gels restored its original shape, yet the recovery rate was noticeably different by the choice of metal ions (Na$^+$ - 90s, Ca$^{2+}$ - 72s, Al$^{3+}$ - 30s). The different speed arises from the differences in relaxation times. As shown in Fig. 3a, the rate of ion-pairing and polymer relaxation differ by metal valency, which affects the driving force for chains to recover their original conformation. During 'programming' at elevated temperatures, the slow relaxation of Al$^{3+}$ ions results in gels that are far from equilibrium, driving faster recovery. The effect of pH was observed from the different degrees of swelling of MPEC gels in various solvent conditions. A bilayer beam of Na$^+$-MPECs with low-pH (left) and high-pH (right), partially crosslinked with MBAA to prevent gel dissolution, changes the curvature depending on the types of solvents submerged into. The larger swelling degree of higher-pH gels in water leads the construct to bend towards low-pH gel (left) whereas the beam switches the bending direction towards high-pH gel (right) in metal solution. The higher propensity of metal ions binding for higher-pH gels induces lower degree of swelling of gels in metal solution.

In summary, we present the design, characterization, and multi-scale modeling of additively manufactured metallo−polyelectrolyte gels. The advanced characterizations through the lens of thermodynamics and mechanics demonstrate the remarkable interplay of dynamic crosslinker valency and polyanion charge sparsity in governing material properties of MPEC gels. The use of theoretical and computational approaches led to a fundamental understanding of the experimental findings and provided insights into the underlying mechanisms from local bond relaxation to mesoscale polymer network topology. The theory developed for this work is capable of extension to other material systems using the key chemical levers identified/explored herein. The development of a simple synthesis pathway and integration to additive manufacturing offers a large parameter space for material design of MPEC gels and an abundance of applications of metallo-polyelectrolytes from upcycling of plastics, predictable mechanical deformation, to stimuli-responsiveness, while opening new avenues to exploit dynamic bonding for various advanced functional applications.

## Methods

### 3D printable photoresin preparation

15 mL of acrylic acid (Sigma Aldrich, >99%) and 2.176 mL of 2 M sodium acrylate (Sigma Aldrich) were mixed to prepare a co-monomer solution. To facilitate comparison between the different systems, the metal ion to sodium acrylate mole ratio was kept at a constant 1:3 under constant molarity of the monomer (8.8 M). Different ionic strength of metal cations was accounted by adding the solution of 0.726 mL of 6 M, 3 M, 2 M metal nitrate (Sigma Aldrich) for mono-, di-, and tri-metal ions, respectively. 2.4 mL of Mili-Q H$_2$O, 2 M or 4 M sodium hydroxide (NaOH) mixture were used as a base to control the different pH range of the resin. To produce an environmentally-stable and high longevity material, 2.5 mL of glycerol (Sigma Aldrich) was then added to the solution as a co-solvent. To obtain accurate pH measurement, pH of resin was measured before adding glycerol. Separately, the

photopackage consisted of 0.632 mL of ethyl(2,4,6-tri-methylbenzoyl) phenylphosphinate (TPO-L) photoinitiator (Oakwood Chemical) stirred into 1.371 mL of dimethylformamide (DMF) (Sigma Aldrich). This solution was then added to the aqueous co-monomer/metal nitrate mixture and swirled until it completely became homogeneous. For metal ion that do not show appreciable UV-Visible absorbance (greater than solvent baseline) at initiation wavelength of 405 nm, 1v/v% of 30mg/mL tartrazine (Sigma Aldrich) was used as a photoblocker without further purification.

### LCD 3D printing

The formulated resin was printed into a 3D structure using a commercial 405 nm wavelength monochrome LCD mSLA printer (Mars 2 Pro, Elegoo Inc.). 50 $\mu$m layer thickness was exposed for 30 s with the laser power 1.99 mW/cm$^2$. The detailed printing set-up and parameters were provided in Supplementary Fig. 1 and Supplementary Table 2.

### Sample post-processing

To remove volatile unreacted photoresin precursors, the printed samples were placed on a PTFE (Teflon©) sheet and dried for 24 h at room temperature in a vacuum oven. Then, the samples were equilibrated at the ambient environment (22.5 °C, 45% RH, 1 atm) for 48 h prior to any characterization. The equilibration time was rigorously determined by way of a dehydration study to ensure consistent mass stability under laboratory environmental conditions. Supplementary Fig. 2 shows that two (2) full days offered consistency across the range of tested metal cations for printed sample volumes and geometries with samples maintaining consistent mass at ~85−90% of initial as-printed part mass.

### ATR-FTIR spectroscopy

ATR-FTIR data was obtained for MPEC samples using the Nicolet iS50-FTIR (Thermoscientific) and the Dura Scope (SensIR Technologies). A DTGS KBr detector and a KBr beam splitter were used to perform 256 scans from 600−4000 cm$^{-1}$ with a unique background correction taken before each scan.

### X-ray Fluorescence Microscope (XRF) imaging

Dehydrated sample elemental distribution mappings for metal centers to determine uniformity and homogeneity were conducted using X-Ray Fluorescence Spectroscopy (Micro-XRF M4 TORNADO Spectrometer, Bruker). Mapping-mode was used for identical sample area (3.5 mm × 3.5 mm), sample thickness (2.5 mm), set resolution (<20 $\mu$m), and dwell time (40 ms/px) under 50 kV, 600 $\mu$A conditions. Spectral counts were reported directly, see Supplementary Information for details.

### Tensile testing

Uniaxial tensile properties of MPEC gels were measured using Universal Testing Machine (Instron, 500 N load cell) with a loading rate of $\dot{\epsilon} \approx 0.03$ s$^{-1}$. Dog-bone shaped specimens with ASTM-D638V standards were used to measure the load and displacement. To prevent the slip of the sample at high strain, 120 grit sand paper sheet was adhered to the grip and the sample.

### Fracture testing

Fracture testing was conducted following methods for the pure shear test for polymers devised by Thomas, Rivlin, and Lake[30]. MPEC samples with the dimensions of 40 mm × 26 mm × 1.5 mm were fabricated using vat polymerization. An initial notch length of 16 mm ($a_0/W = 0.4$) was introduced using a razor blade. To prevent self-healing and preserve sharpness at the notch-tip, a teflon sheet was inserted at the crack and removed immediately prior to the fracture testing. Paired notched and un-notched samples with a gauge length ($H_0$) of 10 mm were tested in uniaxial plane-stress using a material testing system (MTS Systems Co.,

Eden Prairie, MN) at a strain rate of 0.02 s⁻¹. To prevent the slip of the sample at high strain, 120 grit sand paper sheet was adhered to the grip and the sample. Samples were speckled with black spray-paint on top of a thin layer of white spray-paint (Painter's Touch® flat white/black, Rust-oleum) to provide markers for subsequent digital image correlation (DIC) analysis, described below. Imaging was performed using a CCD camera (IL4, Fastec Imaging Coorp., San Diego, CA) at 24 frames-per-second. For the notched sample, the critical strain $\varepsilon_c$ was taken as the global strain at the first point of fracture (notch extension) to establish the most conservative fracture energy case. Fracture energy was calculated using the initial gauge length, the un-notched sample stress-strain curve, and $\varepsilon_c$ as per ref. [38].

### Digital Image Correlation (DIC) analysis
DIC analysis was performed to monitor gel slippage and capture $\varepsilon_c$ using a commercial software, VIC-2D (Correlated Solutions, Columbia, SC)[39]. An incremental correlation methodology with a subset size of $15 \times 15$ px² and a step size of 1 was implemented in the analysis. To mitigate sharp discontinuities in the calculated strain fields, a strain filtering radius of 15 px is applied[39].

### Dynamic Mechanical Analysis (DMA)
DMA (DMA 850, TA Instruments) was used to characterize the thermo-mechanical properties of MPEC gels. For thermal phase characterizations, rectangular shaped specimens with a length of 30 mm, width of 6 mm and thickness of 1.5 mm were heated in the temperature range −50–220 °C with the heating rate of 5 °C min⁻¹. The samples were imposed with a pre-load of 0.2 N and an oscillatory strain of 0.1% at a frequency of 1 Hz. Storage modulus ($E'$), loss modulus ($E''$) and loss factor ($\tan\delta$) of the samples were measured over the experiment duration.

For stress relaxation testing, dog-bone shaped specimens with a length of 32 mm, width of 1.6 mm and gauge length of 4.765 mm were used. The samples were stretched under a strain of 200% in 0.1s and hold at the fixed strain for 1hr at the ambient condition. The relaxed stress and modulus of samples were measured over the experiment duration. To probe the recovery capability of the MPEC samples, displacement change was also tracked instantly after the release of the 1hr hold at the fixed strain. (Due to the incomplete rubbery regime for water evaporation, an effective plateau modulus was obtained from the storage modulus at $T_g \ll T \sim 150\,°C \leq T_{exo-endo}$).

### Differential Scanning Calorimetry (DSC)
DSC (DSC 25, TA Instruments) was used to characterize thermal behavior of MPEC gels. Samples of $\approx 5 \pm 1$ mg were heated −40–220 °C at a ramp rate of 5 °C min⁻¹ under 25 ml min⁻¹ air flow. Both derivative-extremization and mid-point methods for determination of glass transition were employed for each transition regime in accordance with ISO 11357 and ASTM E1356. To probe the existence of the water solvation shell in the MPEC gels, samples were cycled from −40–210 °C and back three times (3x). Heat flow to the samples was measured over the experiment duration with extrema evaluated using the heat flux derivative with respect to temperature.

### Thermal Gravimetric Analysis (TGA)
TGA (TGA 550A, TA Instruments) was used to characterize weight loss profile of MPEC gels. All samples were heated from 20–1000 °C at a ramp rate of 5 °C min⁻¹ under 25 ml min⁻¹ air flow. Mass of the samples were measured over the experiment duration. The weight loss derivative with respect to temperature was used to identify regions of interest in relation to heat flow signatures observed in DSC.

### Confocal microscope imaging
3D-printed thin films of MPEC with the dimensions of 3 mm x 3 mm, were imaged via the inverted laser scanning confocal microscope

(LSM 800, Zeiss). A 640 laser line was scanned with the speed of 5 scans per second across 800 μm by 800 μm along length and width in the center of the thin samples. All samples were measured to a depth of 250 μm. Cross-sections of the z-stack for comparison were taken at a depth of 50±25 μm using 3 GaAsP detectors. Poly(acrylic acid) is autofluorescent (purple) at 640 nm where negative void space (black) corresponds to fluid filled regions, enabling determination of the configuration and distribution of polymer-rich regions of the MPEC gels.

### Scanning Electron Microscope (SEM) imaging
The surface and interior morphologies of the MPEC samples were imaged via SEM (FEI Versa 3D DualBeam) at an operating voltage of 20 kV and a beam current of 5–30 pA using conductive copper tape for stub adhesion. Operating voltage and beam current were adjusted based on the charge sparsity and metal cation in the respective gel to minimize e-beam degradation of the hydrated gels. All gels were measured in-situ to observe the real morphology, which indicates the gel still contains water-glycerol solvent in the gel.

### Gel swelling/dissolution testing
Swelling and dissolution performance of MPEC gels with different metal cations and pH were recorded. $70 \pm 8$ mg of the material was immersed in 10 ml of milliQ© H₂O for the targeted time interval at the room temperature. To maintain the closed system conditions, vials were wrapped with para-film and stored in a steady environment at 22.5 °C without agitation. After immersion for the set time, solid pieces were extracted and weighted. The mass change of samples over the dissolution time were tracked by calculating $\Delta m\% = (m(t) - m_i)/(m_i) \times 100$ where $m_i$ is the initial mass of equilibrated MPEC and $m(t)$ is the mass of current samples.

### Reprocessability
3D-printed MPECs gels were dissolved in H₂O at 80 °C for 24 h. The dissolved polymer melt was poured into a silicone mold and cured into a new shape at 80 °C overnight.

### Shape memory properties
A flower-shaped MPEC gel with open petals was heated to 90 °C and deformed ('programming') to a closed shape for 5 s. Then, the gel was cooled down to −5 °C to fix it to the programmed shape. To restore its original shape and demonstrate a blooming flower, the gel with the programmed shape was heated to 90 °C.

### Bilayer construct of MPECs
A bilayer construct of MPECs was fabricated by exchanging the resin bath during the printing. After the first part of the construct was printed, the new resin was introduced and the air-jet was applied to blow away the precursor solution residuals on the printed part and minimize the material contamination during the exchange. To prevent the diffusion degree of chemical species at the interface of two materials, the printed construct was quickly vacuum dried and post-processed.

### Density Functional Theory (DFT)
The quantum chemistry calculations performed in the work were carried-out using the ORCA package[40]. Geometry optimization was performed using the B3LYP functional[41] with the def2-TZVPP basis set[42,43] and Grimme's D3 dispersion correction[44]. The CPCM solvation model[45] was used to model the aqueous environment. The geometry optimization was performed using the default convergence criteria in ORCA. The optimization was performed using the ORCA 5.0.1 binary. The input structure was generated using Avogadro[46]. Similar scripts were used for the optimization of the acrylate monomers in the absence of metal ions.

## Molecular Dynamics (MD) simulations

The MD simulations of the gel systems were carried-out using the Large-scale Atomic/Molecular Massively Parallel Simulator (LAMMPS)[47]. The Kremer–Grest[48] coarse-grained force-field was used to model the polymer chains. All simulations were run with 50 polyelectrolyte chains each with 300 beads. An implicit solvent model was used where the effects of the solvents were accounted for using the system dielectric constant and pressure. This effectively models all solvents present as just two parameters. Following energy minimization and equilibriation in an *NpT* ensemble, two production simulations were performed: an *NVE* simulation to analyse the relaxation modes and a non-equilibrium simulation with elastic-deformation in the *x* plane.

## Mean-field theory

Following the theory developed in prior works[49,50], we developed an expression for free energy of a sparsely-charged polyelectrolyte system in the presence multivalent ions. Similar to the molecular dynamics simulations, an implicit solvent model was used where the effects of the solvent were accounted for using the system dielectric constant. The detailed derivation of the theory is provided within the supplementary information. The free-energy expression was implemented using the Clapeyron.jl[51] package.

## Data availability

Data generated and/or analysed during this study are included in the Supplementary Information. Further data are available from the corresponding authors on request.

## Code availability

Example codes used to perform DFT, MD and mean-field theory calculations are provided on GitHub: https://github.com/zgw-group/Metallo-Polyelectrolyte-Gels[52].

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

## Acknowledgements

The authors gratefully acknowledge the financial support from the Center for Autonomous Systems and Technologies (CAST) at Caltech and the Schwartz/Reisman Collaborative Science Program from the Schwartz/Reisman Foundation. Research was also sponsored, in part, by the U.S. Army Research Office and accomplished under contract W911NF-19-D-0001 for the Institute for Collaborative Biotechnologies. We thank Beckman Institute as well as the critical support of Molecular Materials Research Center (MMRC), Beckman Bioimaging Facility (BIF), Beckman X-Ray Crystallography Facility (XRCF), and Geological and Planetary Sciences Division Analytical Facility at Caltech. Z-G.W. acknowledges funding from Hong Kong Quantum AI Lab, AIRinnoHK of the Hong Kong Government. We also acknowledge Dr. Daryl Yee and Dr. Widianto Moestopo for their contribution to project ideation and Sophie Howell for supporting preliminary feasibility studies. We thank Dr. Chi Ma, Dr. Giada Spigolon, Dr. Kathy Faber and Dr. Guruswami Ravichandran at Caltech for equipment support and technical assistance in this research. Useful discussions with Sammy Shaker, Alec Glisman, Alexandros Tsamopoulos, Dr. Alejandro Gallegos, Dr. George Rossman, and Dr. Igor Lubomirsky are also gratefully acknowledged.

## Author contributions

S.L., Z-G.W. and J.R.G. conceived the project, initiated collaborations, and coordinated conceptual design for the project. A.C. and Z.W.T. developed the initial photoresin formulation and selected solution dielectric for gel. S.L. optimized the fabrication protocols for multivalent and pH-tunable system. S.L. and S.J.V. prepared most samples for characterizations. S.J.V. developed the understanding of and designed experiments related to the effect of photoresin pH on charge sparsity in the polyelectrolyte gels. S.L. carried out most of mechanical/thermo-mechanical characterizations. C.F., V.G. and S.L. designed and performed fracture study. S.J.V. contributed most of spectroscopy, microscopy, and thermal-kinetic characterization. P.J.W. devised and performed the DFT calculations and the Molecular Dynamics simulations. P.J.W., S.J.V. and Z-G.W. derived a modified Henderson-Hasselbalch model. P.J.W. and Z-G.W derived the mean-field associative polyelectrolyte theory and generated the phase diagrams. S.L. and S.J.V. designed and performed gel dissolution, swelling, and application experiments. S.L., S.J.V., P.J.W., Z-G.W, and J.R.G. discussed & interpreted experiments and theoretical results collaboratively, and together wrote the manuscript. All authors edited and approved the manuscript.

## Competing interests

The authors declare no competing interests.
