## [Peer Review File · Nature Communications]

Molecular Control via Dynamic Bonding Enables Material Responsiveness in Additively Manufactured Metallo-PolyelectrolytesReviewers' Comments:

Reviewer #1:

Remarks to the Author:

Lee and coworkers have presented an additively manufactured gel system leveraging COO⁻/M⁺ non-covalent interactions. The mechanical properties of these polyacrylic acid (PAA)-based metal ion-coordinated gels were characterized, with adjustments to metal ion valency and electrolyte charge density achieved by pH modification. An associative mean-field theory underpinned the phase behavior analysis and subsequent phase diagram construction. While the theoretical aspects of this work, integrating density functional theory, molecular dynamics, and mean-field theory, are robust and potentially extensible to other ionic gel systems, I have reservations about the experimental approach and results.

First, the experiments revealed the fact that the mechanical strength is related to the charge density and metal ion valency, both of which were already known, qualitatively. It will be highly desirable to “quantify” the influence and compare with the theoretical calculations.

Second, the gel system is complicated in terms of the number of components involved and may not be the best model system to elucidate the underlying physics and compare with the theoretical calculations.

The manuscript needs to be carefully revised before it can be considered publication in Nature Communications. I propose three major questions for the authors to address:

(1) Control parameters: The claim of two critical control parameters—metal ion valency and charge sparsity—within a system comprising seven components is ambitious. AA units, SA units, pH modifier (nitric acid or NaOH), nitrate salt, water, glycerol, DMF. With so many degrees of freedom (although the concentrations of some were fixed), it is very hard to completely decouple the influence of each individual component on the macroscopic properties. The most apparent one is the role of solvent. The mechanical strength of the gel can be highly dependent on the weight fraction and composition of the solvent of the gel which were not provided in the manuscript. The authors should comment on the role of solvent, especially when the theoretical calculation (e.g., Fig 4a) was done in water, not the ternary co-solvent system.

(2) Thermodynamic Equilibrium: The assertion that the sample characterization elucidates the assembly's thermodynamics is precarious. The synthesis process, a one-pot photopolymerization within a 3D printer, followed by evaporation of monomers under ambient condition, casts doubt on whether the network achieved thermodynamic equilibrium—something that is not typically reached without a controlled thermal treatment. Thermal analyses (Fig 2b and 2c) may not reveal the actual properties. The involvement of three different solvents worsens the situation.

The authors should comment on the thermal stability of the samples and whether the reported properties represent the equilibrium state. Fig 4c serves as a good example here: what are the properties of the re-casted samples compared to the printed ones?

(3) Cluster Formation: The authors mentioned the clusters formation in these electrolyte gels. It has been widely accepted that the amount and size of clusters in such ion-containing complex systems dominate the polymer dynamics and mechanical properties. The clustering effect may be the third critical parameter (besides charge density and valency) and needs to be studied in more detail experimentally, e.g., what is the morphology, size, fraction of such clusters? X-ray scattering may be a powerful tool.

Other comments for the authors to consider:

(1) Why is a small amount of sodium acrylate needed, given the fact that the acrylic acid units will be converted to sodium acrylate when pH is adjusted?

(2) The secondary T_g in Fig. 3c is questionable since the thermal history was not removed.

(3) The exotherm at 140 to 170 C was ascribed to water evaporation, could it also be DMF evaporation?

(4) Fig 2d demonstrates that the Al³⁺ system is better than the covalent system. Please comment on the solvent content and the contribution of clusters in both cases.

- (5) More experimental data points will be helpful to validate the predicted phase diagrams (Fig 4a and 4b). For direct comparison, the solvent should be kept consistent with the theoretical calculations (water?).
- (6) Reconsider the term "upcycling" in Fig 4c to a more precise term such as "solvent reprocessing".
- (7) Many figures are too small and almost impossible to read.

Reviewer #2:

Remarks to the Author:

This manuscript demonstrates a robust and long-life MPEC gel prepared by stereolithography-based additive technology. Combined with experimental results and computational simulation, the authors comprehensively elucidate the significant impacts of two microscale factors, metal ion valency and polyelectrolyte charge sparsity, on various macroscopic properties of MPEC gel materials. It bridges the molecular-level chemistry of dynamic bonds and the continuum-level material properties, concurrently paving the way for innovative applications of dynamic bonding. The results and conclusions can deepen the understanding of the relationship between local molecular-scale interactions and global properties of MPEC gels. The methodology, the integrity of the data, and the quality of presentation are compelling. I think it is a nice work and deserves publication in this journal. Other comments and suggestions are as follows.

1. What is the water (or mixture solvent) content of the gels? The mechanical properties and coordination cluster density severely depend on the water content. These values for different gels should be provided.
2. Since the gels are synthesized from a highly concentrated pregel solutions, entanglements should contribute a lot to the viscoelastic and mechanical properties. Thus, it should be better to discuss this influence in this manuscript.
3. The coordinate bond strength is affected by the charged valence of the metal ion, but it also depends on metallic ions. For example, the Ca^{2+} , Zn^{2+} , and Fe^{2+} have different coordination bond strengths with carboxyl groups. Could you give some discussions on this point based on theoretical analysis? In addition, How about the case of tetravalent coordinating ions such as Zr(IV) and Ti(IV) , besides Na^+ , Ca^{2+} and Fe^{3+} ?
4. The structural mode of complexation of metal species is crucial, while the bidentate chelation relationship between each metal ion and the polymer has been elucidated through FTIR-ATR recordings referred to the Ref 20 and Ref 21. Are there any other characterization methods available to provide additional evidence for this conclusion?
5. In Fig. 2b, the DMA and DSC results indicate a glassy-rubbery transition. If this is a gel, what is the water content? Why is this gel in a glassy state at relatively low temperatures? The Young's modulus in Fig. 2d of the gel is several MPa, much lower than the storage modulus in Fig. 2b. What's the reason?
6. It is very meaningful that MD results can distinct the cluster size of the gels with different metal ions. What's the reason for the formation of cluster, and which factor determines the size? What's the relation of the strength between clusters and single coordinate bond?
7. Fig. 4d shows interesting bending behavior of the gel at different strain rate. This result is explained in terms of viscoelasticity of the gel with dense physical associations. It should be better to study the rheological behavior and correlate with the mechanical performance of the gel.

Reviewer #3:

Remarks to the Author:

The authors prepare an aqueous solution of monomers, initiator, and ions. They vary pH and concentration of ions of different valencies (Na^{1+} , Ca^{2+} , and Al^{3+}) in the precursor. The authors then use Liquid Crystal Display stereolithography, through radical polymerization, to fabricate hydrogels. These hydrogels are crosslinked through ionic interactions. They characterize the hydrogels in many ways (stress-strain curves, toughness, Fourier transform infrared spectroscopy, X-ray fluorescence Microscopy, differential scanning calorimetry, optical microscopy, thermal gravimetric analysis, dynamic mechanical analysis, scanning electron microscopy, and confocal microscopy). They conduct numerical simulations to predict the strength of ionic interactions of crosslink and interchain/intrachain crosslink ratio of network topology. The authors also construct phase diagrams to help identify the required synthesis conditions to fabricate tough hydrogels.

The combination of experiments of many varieties, as well as calculations, makes this work unusually comprehensive among works on the mechanical behavior of hydrogels. The experiments and calculations shed insight on ionically crosslinked polymer hydrogels. Furthermore, the method of fabrication enables the production of structures of complex shapes. The authors demonstrate the gel recyclability, mechanical response of bilayer structure, and the morphing of the structures in response to temperature and solvent.

The ionically crosslinked polymer hydrogels have been studied extensively. In the abstract, the authors state "We discover that the mono-, di-, and trivalent metal ions afford control of the coordination environment and bond strength within the polymer matrix, which propagates to the macroscale properties where higher valency ions result in stiffer and tougher materials." It is unclear which part is a discovery. The effect of multivalent ions on mechanical properties of polyelectrolytes have been studied in an enormous number of papers. Some examples are listed below. In light of these papers and many other papers in the literature on the ionically crosslinked polymer hydrogels, the authors need to review this literature and make a clear statement of the novelty of the paper.

1. Zhong, Ming, et al. "Dually cross-linked single network poly (acrylic acid) hydrogels with superior mechanical properties and water absorbency." *Soft matter* 12.24 (2016): 5420-5428.
2. Gao, Yang, et al. "Photodetachable adhesion." *Advanced Materials* 31.6 (2019): 1806948.
3. Dong, Hong, et al. "Cation-induced hydrogels of cellulose nanofibrils with tunable moduli." *Biomacromolecules* 14.9 (2013): 3338-3345.
4. Zhu, Qian, et al. "Effect of ionic crosslinking on the swelling and mechanical response of model superabsorbent polymer hydrogels for internally cured concrete." *Materials and Structures* 48 (2015): 2261-2276.
5. Mørch, Yrr A., et al. "Effect of Ca^{2+} , Ba^{2+} , and Sr^{2+} on alginate microbeads." *Biomacromolecules* 7.5 (2006): 1471-1480.
6. Baumberger, Tristan, et al. "Cooperative effect of stress and ion displacement on the dynamics of cross-link unzipping and rupture of alginate gels." *Biomacromolecules* 11.6 (2010): 1571-1578.

Response to Reviewers

We would like to thank the reviewers for providing insightful feedback. We are delighted that both found the manuscript interesting. We have addressed all of their comments below and have highlighted any of the changes to the manuscript in red within the main body.

Reviewer 1		
Number	Comment	Response
0	Lee and coworkers have presented an additively manufactured gel system leveraging COO-/M+ non-covalent interactions. The mechanical properties of these polyacrylic acid (PAA)-based metal ion-coordinated gels were characterized, with adjustments to metal ion valency and electrolyte charge density achieved by pH modification. An associative mean-field theory underpinned the phase behavior analysis and subsequent phase diagram construction. While the theoretical aspects of this work, integrating density functional theory, molecular dynamics, and mean-field theory, are robust and potentially extensible to other ionic gel systems, I have reservations about the experimental approach and results. First, the experiments revealed the fact that the mechanical strength is related to the charge density and metal ion valency, both of which were already known, qualitatively. It will be highly desirable to “quantify” the influence and compare with the theoretical calculations. Second, the gel system is complicated in terms of the number of components involved and may not be the best model system to elucidate the underlying physics and compare with the theoretical calculations. The manuscript needs to be carefully revised before it can be considered publication in Nature Communications. I propose three major questions for the authors to address:	We are grateful to the reviewer for the constructive criticism, which has provided us with valuable insights for improving our paper. In response to the reviewer’s suggestions, we performed a significant number of experiments to thoroughly investigate the issues related to thermal equilibrium, solvent effects, and phase behavior.

1 Control parameters: The claim of two critical control parameters—metal ion valency and charge sparsity—within a system comprising seven components is ambitious. AA units, SA units, pH modifier (nitric acid or NaOH), nitrate salt, water, glycerol, DMF. With so many degrees of freedom (although the concentrations of some were fixed), it is very hard to completely decouple the influence of each individual component on the macroscopic properties. The most apparent one is the role of solvent. The mechanical strength of the gel can be highly dependent on the weight fraction and composition of the solvent of the gel which were not provided in the manuscript. The authors should comment on the role of solvent, especially when the theoretical calculation (e.g., Fig 4a) was done in water, not the ternary co-solvent system.

We greatly appreciate the reviewer’s insightful comments regarding the complexity of our system and the challenges in isolating the effects of individual components. Indeed, with seven components involved, it is challenging to completely decouple the influence of each component on the macroscopic properties. To address this concern, we made concerted efforts to maintain consistency in various parameters during the material design process, allowing for a more controlled comparison.

Specifically, we ensured that parameters such as monomer type and concentrations (AA and SA units), co-ion identity (nitrate ions), solvent quantity (water, glycerol, and DMF), and polymerization conditions (exposure conditions) were kept constant across all experiments. After printing the gels, we conducted Thermogravimetric Analysis (TGA) to quantify solvent quantities. TGA solvent analysis of loosely bound water (hydrating) and tightly bound water (solvating) shows that, there is a negligible variations in water content for gels synthesized with different pH and metal species. On average, gels possess $\sim 14\% \pm 2.14\%$ water by weight with a median of 14.27% water across all samples, from those without metal to trivalent and from pH 0.9 to 3.25. **Details on these additional experiments are provided in Section 6 of the revised SI.**

With these efforts and verifications, we believe that the variations in metal ion valency and system pH, which we identified as having the most significant impact on material properties, could be reasonably isolated for comparison. To explicitly address this concern, we included the following statements in the main text (Section 2) of the revised manuscript: “For fair comparison, most of synthesis parameters, except the metal valency and system pH, were kept consistent between the gels.” and “Themogravimetric analysis (TGA) confirms that all equilibrated gels, irrespective of the metal ions and the system pH chosen, contain $\sim 14\% \pm 2.14\%$ water by weight. This uniformity underscores the consistency in solvent content across all samples.”

In terms of theoretical calculations, we employed an implicit solvent model for both MD simulations and theoretical studies to capture the experimental ternary co-solvent systems. While this model simplifies the ternary co-solvent to just two defining parameters, the dielectric constant and free space of the system, it is intended to capture the overall influence of the solvents. This approach is a common approximation in the field due to computational feasibility and the limited additional impact of explicit solvent modeling, beyond these two parameters.

We appreciate the reviewer’s recognition of the significance of solvent effects on material behavior. To address this point, we conducted additional thorough studies on the effect of solvent using both experimental and computational approaches. **We have included a new section in the revised SI (Section 7) discussing the effects of solvent on mechanical properties.** While we recognize the importance of solvent control, we decided to explore this aspect in the SI to avoid substantially lengthening the main manuscript. A summary of the key findings from this section includes:

- Increasing water content from 10% to 20% and 30% weight results in a reduction of modulus by a factor of 1.41 and 4.80 for Al-MPEC and 1.86 and 3.92 for Ca-MPEC, as observed in tensile tests.
 - The increase in water content also leads to a significant reduction in hysteresis, observed during the tensile test for all gels.
 - MD simulations indicate that an increase in free volume reduces the polymer relaxation time while ion-pairing relaxation is less affected.
 - Faster polymer relaxation leads to more dynamic clusters and a wider distribution of cluster sizes. However, trivalent metal ion systems are less affected due to the strong electrostatic interactions.
 - The one-pot synthesis platform allows for tuning with different solvents, and tuning glycerol content produces similar effects to water.
-

2 Thermodynamic Equilibrium: The assertion that the sample characterization elucidates the assembly's thermodynamics is precarious. The synthesis process, a one-pot photopolymerization within a 3D printer, followed by evaporation of monomers under ambient condition, casts doubt on whether the network achieved thermodynamic equilibrium—something that is not typically reached without a controlled thermal treatment. Thermal analyses (Fig 2b and 2c) may not reveal the actual properties. The involvement of three different solvents worsens the situation. The authors should comment on the thermal stability of the samples and whether the reported properties represent the equilibrium state. Fig 4c serves as a good example here: what are the properties of the re-casted samples compared to the printed ones?

We are grateful to the reviewer for raising this critical point. Understanding whether our gels achieved thermodynamic equilibrium is essential for interpreting our results and comparing them with theoretical predictions. To address this concern, we conducted additional experiments and found multiple lines of evidence supporting the equilibrium state of our gels.

Firstly, during the gel synthesis process, we tracked the mass of the gels over fabrication stages and observed that they reached equilibrium densities within a three-day after post-processing. Reintroduction of water after equilibrium by hydrating the gels in a high-humidity environment (RH 80%) ensures the gels undergoes mass increase due to hydration and then always return back to the original mass after a certain periods of time in the environment. This provided confidence that the solvents had likely reached their equilibrium state.

Fig. R1 Mass tracking of MPEC-gels over fabrication stages. The current mass is normalized by the initial mass to measure the mass change.

Secondly, to assess the repeatability of thermal signatures exhibited by the gels, we subjected them to multiple cycles of heating and cooling between room temperature and 60°C with a rate of 5°C/min at a fixed frequency (1 Hz). By isolating the effects of solvent evaporation on the measured storage and tan delta values, we found consistent thermal signatures over more than five cycles for each gel.

Fig. R2 Thermal cycles over 25°C to 60°C and the subsequent frequency response after equilibrated at 60°C for **a,c** Ca²⁺ and **b,d** Al³⁺-MPEC gels .

Different line transparency represents the response from different cycles. Gels were re-hydrated and re-equilibrated in ambient conditions before subsequent cycle to ensure comparable solvent degree.

Lastly, to validate the reproducible response of the gels at higher temperatures well beyond the glass transition temperature (T_g), we extended our analysis to 110°C. We took precautions to ensure equitable comparisons between cycles by re-solvating the tested gels in a high-humidity environment (RH 80%) for over 24 hours before each subsequent cycle. To avoid the thermal drift from the temperature ramping, the gels were equilibrated at 40°C, 60°C, 80°C, and 100°C for 10 minutes during the temperature ramp. Then, rheological measurements, sweeping the frequency from 0.1 Hz to 100 Hz, were conducted at each equilibrated temperature. The results confirmed the consistency of thermal signatures across all cycles, even at elevated temperatures.

Fig. R3 **a** Mass tracking of Ca²⁺ (green) and Al³⁺-MPEC gels (yellow) to ensure constant solvent content between tests. **b** Temperature profile applied during the test, using four different equilibration temperatures to probe the rheological behavior of **c** Ca²⁺ (green) and **d** Al³⁺-MPEC gels.

These findings collectively support the assertion that the gels achieved thermodynamic equilibrium, providing a solid foundation for interpreting our results and comparing them with theoretical predictions.

Cluster Formation: The authors mentioned the clusters formation in these electrolyte gels. It has been widely accepted that the amount and size of clusters in such ion-containing complex systems dominate the polymer dynamics and mechanical properties. The clustering effect may be the third critical parameter (besides charge density and valency) and needs to be studied in more detail experimentally, e.g., what is the morphology, size, fraction of such clusters? X-ray scattering may be a powerful tool.

This is indeed an important aspect, and we appreciate the reviewer’s attention to this detail. We acknowledge that cluster size and amount play a crucial role in determining the material properties of MPEC gels. However, it’s essential to clarify that in our manuscript, clusters are defined as the percolating network formed by dynamic crosslinks between chains, rather than “ion aggregation/clustering” resulting from strong local electrostatic interactions. We provided detailed elaboration on this distinction within our methods Section.

Given this definition of clustering as the percolating network, measuring cluster size using X-ray scattering techniques poses a challenge due to their larger size, typically in the range of micrometers. From our MD simulations, we observed that the clusters span almost the entire gel, indicating substantial gel network formation. This observation is further supported by confocal microscope images provided in the Supplementary Fig. 9, demonstrating a large, continuous span of polymer network across the sample size. The estimated size of these clusters exceeds the maximum size scale detectable by X-ray scattering techniques, which typically range up to tens of nanometers. As shown in the figure below, X-ray scattering signatures from our simulations did not reveal significant peaks at low structural factor (q), and experimental observations also align with this result.

Simulations:

Fig. R3 Simulated X-ray scattering signatures for mono- (blue), di- (green), and trivalent (yellow) gels.

Experiments:

Fig. R4 Experimentally measured SAXS spectrum for Na^+ (blue), Ca^{2+} (green), and Al^{3+} (yellow) gels.

-
- 4 Why is a small amount of sodium acrylate needed, given the fact that the acrylic acid units will be converted to sodium acrylate when pH is adjusted? We thank the reviewer for this inquiry and they are indeed correct. However, the addition of a small quantity of sodium acrylate serves as a buffer to facilitate common ion effects. Since the system contains a significant amount of acrylic acid (strongly acidic), manually adjusting the pH using a strong buffer can be challenging. The inclusion of sodium acrylate aids in pH adjustment, simplifying the process for experimental convenience.
-
- 5 The secondary Tg in Fig. 3c is questionable since the thermos history was not removed. As mentioned earlier in comment #3 regarding thermal equilibrium, we observed consistent thermal signatures for low pH gels. With insights from MD simulations suggesting a reduction in relaxation time for higher pH systems, we strongly believe that the system has reached thermal equilibrium. As such, we maintain that the secondary T_g is indeed real.
-

6 The exotherm at 140 to 170 C was ascribed to water evaporation, could it also be DMF evaporation?

Thank you for the insightful comment. This is an excellent question given the low vapour pressure of DMF 0.9-19 kPa from 300-370 K (J. Chem. Eng. Data 2006, 51, 5, 1860–1861) in addition to its boiling point at 426.15 K (Encyclopedia of Reagents for Organic Synthesis, 2001). Nevertheless, the low mass loading of DMF in the system ($\ll 5\%$) together with the vacuum pumping for 24 hours before equilibration at 30 mmHg is believed to have removed all residual DMF. We have performed three analyses to corroborate our hypothesis that these thermal signatures are related to water and not DMF: (1) Mass Spectrometry; (2) Kissinger kinetics of dehydration and comparison with energies for dehydration of similar compounds in literature; and (3) reformulating resins to remove the needs for DMF by change of photo-initiator. For the reviewer’s interest, the details of these experiments are described below.

(1) Mass spectrometry (MS) on a equilibrated sample with the largest exotherm-endotherm transition (Aluminum MPEC) performed a Residual Gas Analyzer (RGA 200 system) found no DMF signal. Samples were heated in a screw-capped stainless steel vessel connected by a polyether ether ketone (PEEK) capillary to a HiCube Pfeiffer high-vacuum system from which the RGA sampled the off gas flow. The sample was heated from 300 K to 600 K using a DigiSense Temperature Controller connected to a tie-on resistive heating element and a type-K thermocouple for P.I.D. loop temperature measurement. Samples showed neither a discernible CO_2 signal at 44 a.m.u. nor a DMF signal at 73 a.m.u. from degradation and evaporative loss, respectively. Further, neither set of signals was observed at room temperature, the ramp up, and holding at 600 K. However, as expected, N_2 , O_2 , H_2 gas and water were observed (the limited H_2 gas resulting from the ionization process from water by the RGA filament). The temperature of 600 K was selected to ensure sufficient overheating in case of delayed internal heating of the sample relative to the metal container’s exterior onto which the thermocouple was kapton-taped. 50 scans were taken at each isothermal temperature point to ensure reasonable signal to noise was observed. See attached mass spectrometry data for further information.

Fig. R5 RGA setup and measured mass spectrometry signal for Al^{3+} -MPEC gels. DMF and CO_2 signals were not detected from the sample (indicated arrows).

(2) We performed a kinetic analysis on the this exothermic signature using the Kissinger method consistent with recommendations of the International Confederation for Thermal Analysis and Calorimetry (Thermochim. Acta. 2014;590:1–23; Thermochim. Acta. 2011;520:1–19) using dynamic scanning calorimetry. Scans were performed at differing heating rates and the shift in maximal onset temperature was recorded.

Fig. R6 Kissinger kinetic analysis on the DSC exothermic signatures and extracted activation energies and frequency factors for process.

These experiments determined an average activation energy of ~ 80 - 100 kJ/mol (J. Phys. Chem. C. 2017;121:15392–15401; RSC advances 2.11 (2012): 4664-4674), which is consistent with literature for the dehydration of oxalate compounds (a double-carboxylate metal chelating compound) within a similar temperature regime ($\sim 120 - 160^\circ\text{C}$).

(3) To supplement these necessary data and provide a sufficient evidence conclusively ruling out DMF as the evaporant, we reformulated a DMF-free MPEC for thermal characterization. In short, DMF was only used as a co-solvent owing to the need to make miscible the TPO-L photoinitiator. We replaced the TPO-L photoinitiator, thereby directly removing need of DMF, with lithium phenyl (2,4,6-trimethylbenzoyl) phosphinite (LAP), another free radical photoinitiator with similar absorption wavelengths in the UV-violet range.

Fig. R7 Thermogram for MPEC gels synthesized without DMF solvent. Gels of two different hydration states were run showing reduction in endotherm with removal of solvent (water) under vacuum. This is consistent exotherm-endotherm behavior observed in DMF containing gels within similar temperature ranges, $\sim 120 - 160^\circ\text{C}$. Scale bar 2mm.

Samples showed similar exotherm-endotherm behavior consistent with prior results from dynamic scanning calorimetry. Samples were run at various states of hydration (wet from synthesis) and vacuum dried for 24 hours to confirm endotherm could be controlled with amount of solvent (water) available to be evaporated during heating. This observation was repeatable across multiple samples of various degree of hydration. The reduction in the magnitude of the endotherm after longer vacuum pumping is consistent with our understanding that the residual water is pulled out of the gel. This confirms the observed thermal signature is characteristic of the MPEC gel and not a product of any residual DMF co-solvent that might persist in trace quantities. **This data and discussion has been added to the Supplementary Information and referenced in the main text.**

The range of spectroscopic and thermal evidence supports our claims that vacuum post-processing to remove residual solvent successfully removes those trace quantities of DMF from the system. Our results here are in agreement with literature for similar compounds.

7 Fig 2d demonstrates that the Al³⁺ system is better than the covalent system. Please comment on the solvent content and the contribution of clusters in both cases.

We appreciate the reviewer's comment. As clarified in our response to the first comment, the solvent content of the trivalent gel is comparable to that of the mono-, divalent, and covalent gels, as now stated in the main text. This renders the main differences in mechanical response between the trivalent and covalent systems to be caused by the different polymer network configuration, affected by the size of the crosslinker and the local coordination number. Despite the stronger binding strength in the covalent gel compared to the trivalent cation, the larger size and lower functionality (two, compared to three for Al) of the chemical crosslinker (MBAA) allow more free space between the chains, resulting in a less-dense gel.

The less-dense polymer network of the covalent gels results in an initially less stiff response at smaller strains, as demonstrated in Fig 2d. However, under deformation, the relaxation ability of crosslinkers becomes more significant, leading to a sharp increase in slope as the applied strains increase for the covalent gels. In contrast, the dynamic nature of the Al crosslinks allows them to extend further than the covalent gels, where breaks in the crosslinks are permanent. This mechanism is supported by the reduced stress analysis included in the SI, which confirms that the covalent crosslinker undergoes strain stiffening due to finite chain extensibility, whereas the Al gels undergo strain softening due to the ability of bond dissociation, allowing for chain relaxation.

8 More experimental data points will be helpful to validate the predicted phase diagrams (Fig 4a and 4b). For direct comparison, the solvent should be kept consistent with the theoretical calculations (water?).

We appreciate the reviewer’s helpful suggestion. Quantitatively validating the phase diagrams presented in Figures 4a and 4b proved challenging due to the highly dense phase of the gel in our system. However, we conducted qualitative comparisons to verify the constructed phase diagrams for both Al and Ca. To explore different phase regions on the diagram, equilibrated gels were immersed in various solutions, including water, metal salt solutions with different concentrations, and polyacrylic acid solutions with different concentrations, until equilibrium was reached

Fig. R8 Experimental plan for exploring different phase regions on the constructed phase diagram for Ca^{2+} (left) and Al^{3+} -MPEC (right). The star marker represents an arbitrarily chosen starting gel condition.

Different phases were observed depending on the solution type and concentration, aligning with the behavior predicted from the phase diagram. **We have summarized all results from these tests in Supplementary Figures 14-16 in the SI.** Regarding the solvent consistency, we ensured that the theoretical calculations used implicit solvent parameters to match the experimental conditions, facilitating direct comparison between theory and experiment. We hope this clarifies the approach taken to validate the phase diagrams.

9 Reconsider the term “upcycling” in Fig 4c to a more precise term such as “solvent re-processing”.

Thank you for this suggestion. Upon reflection and a thorough review of the literature (Green Chem. 2021;23;6863-6897; Chem. Eng. Journal. 2023;1385-8947), we agree that ‘reprocessing’ more accurately describes the process in our work. Consequently, we have changed “upcycling” to “reprocessability” in Fig 4c, as this term better reflects the function and application of the material.

10 Many figures are too small and almost impossible to read.

We appreciate the reviewer’s attention to detail. In response, we have increased the font sizes for all figures to enhance readability.

Reviewer 2

Number	Comment	Response
0	This manuscript demonstrates a robust and long-life MPEC gel prepared by stereolithography-based additive technology. Combined with experimental results and computational simulation, the authors comprehensively elucidate the significant impacts of two microscale factors, metal ion valency and polyelectrolyte charge sparsity, on various macroscopic properties of MPEC gel materials. It bridges the molecular-level chemistry of dynamic bonds and the continuum-level material properties, concurrently paving the way for innovative applications of dynamic bonding. The results and conclusions can deepen the understanding of the relationship between local molecular-scale interactions and global properties of MPEC gels. The methodology, the integrity of the data, and the quality of presentation are compelling. I think it is a nice work and deserves publication in this journal. Other comments and suggestions are as follows.	We sincerely appreciate the reviewer's recognition of the potential impact of our work and constructive feedback, which will undoubtedly help us further clarify and strengthen our paper. We have conducted additional experiments and studies to address the reviewer's questions and suggestions.

1 What is the water (or mixture solvent) content of the gels? The mechanical properties and coordination cluster density severely depend on the water content. These values for different gels should be provided.

We thank the reviewer for highlighting the importance of understanding the water content in our gels. In response, we conducted Thermogravimetric Analysis (TGA) to precisely quantify the water content in our equilibrated gels. Across all variations in metal ion types and system pH, our analysis consistently revealed that all gels contain approximately $\sim 14\% \pm 2.14\%$ water by weight with a median of 14.27% water after post-processing. This uniform water content ensures a fair and accurate comparison between gels, enabling us to effectively isolate the effects of valencies and pH. **The detailed results of our TGA analysis, have been tabulated and included in Section 6 in the revised SI for reference.** To clarify the solvent information in the main text, we included a statement in the Section 2: “Thermogravimetric analysis (TGA) confirms that all equilibrated gels, irrespective of the metal ions and the system pH chosen, contain $\sim 14\% \pm 2.14\%$ water by weight. This uniformity underscores the consistency in solvent content across all samples.”

We appreciate the reviewer’s recognition of the significance of solvent effects on material behavior. **In response to this observation, we have added a new Section within the revised SI (Section 7) to thoroughly discuss the effects of solvent on mechanical properties and cluster dynamics.** This discussion is supported by both experimental data and molecular dynamics simulations. As a quick summary of the Section with the key findings:

- Increasing water content from 10% to 20% and 30% weight results in a reduction of modulus by a factor of 1.41 and 4.80 for Al-MPEC and 1.86 and 3.92 for Ca-MPEC, as observed in tensile tests.
- The increase in water content also leads to a significant reduction in hysteresis, observed during the tensile test for all gels.
- MD simulations indicate that an increase in free volume reduces the polymer relaxation time while ion-pairing relaxation is less affected.
- Faster polymer relaxation leads to more dynamic clusters and a wider distribution of cluster sizes. However, trivalent metal ion systems are less affected due to the strong electrostatic interactions.
- The one-pot synthesis platform allows for tuning with different solvents, and tuning glycerol content produces similar effects to water.

2 Since the gels are synthesized from a highly concentrated pregel solutions, entanglements should contribute a lot to the viscoelastic and mechanical properties. Thus, it should be better to discuss this influence in this manuscript.

We value the reviewer’s insightful suggestion regarding the role of entanglements in our gel system. It’s important to note that our gels are synthesized via in-situ radical polymerization of monomeric units, without pre-existing polymer chains in the pregel solution. Thus, entanglement is primarily considered in the gel-state within our study.

To address the influence of entanglements, we treat the monovalent gel as a reference system where entanglement plays a significant role in the material response, given the weaker electrostatic interactions and absence of crosslinking capability. This assumption finds support in our comparison of the effective plateau modulus, which measures the change in storage modulus over temperature at a fixed frequency, between Na-gels and pure PAA-gels formed without any metal ions or crosslinkers (refer to Figure 3b in the main text and Supplementary Fig. 8).

By assuming that time-temperature superposition holds, the consistent rubbery plateau modulus observed between Na and PAA systems, along with the analytically estimated entanglement modulus for purely entangled polymer, suggests that both systems have a similar molecular weight between entanglements (M_e). This finding confirms that the material response in Na gels is predominantly influenced by entanglement effects (Supplementary Fig. 8). However, as highlighted in Figure 3b, the presence of high-valency ions significantly increases the plateau modulus, indicating that while entanglement effects become less dominant, the strong electrostatic interactions of the ions play a more critical role in the material response.

• The effects of entanglement were briefly explained in SI under the plateau modulus analysis. We will explain that, in this study, we are using the Na⁺ system as a representative system for entanglement since the ionic interactions are significantly weaker than other higher-valency metal species.

Fig. R9 Storage modulus of MPECs measured from -20°C to 180°C with the temperature ramp rate of $5^{\circ}\text{C}/\text{min}$. Due to the incomplete rubbery regime for water evaporation, an effective plateau modulus of MPECs was obtained from the storage modulus at $T_g \ll T \sim 150^{\circ}\text{C} \leq T_{exo-endo}$ of the polymer in the rubbery state. Higher valency gels induced higher plateau modulus, deviating from the entanglement modulus limit (G_e) whereas different pH gels did not exhibit noticeable differences. It indicates Na^+ -MPEC represents entanglement-dominant response.

As suggested by the reviewer, we have clarified our consideration of entanglement in the main text by incorporating a sentence in the Section 2: “The weak electrostatic interactions of Na^+ ions allow the monovalent gel to serve as a reference system where entanglement is the major contributor to the material response.”

Building upon the insights gained from considering entanglement, we conducted an extended study by synthesizing Na-gels with varying monomer concentrations. These variations result in different chain densities and lengths, thereby impacting the degree of entanglement within each system. The mechanical data for these gels is provided below. While more careful study is needed to ensure uniform solvent content across all systems, our preliminary findings suggest that the degree of entanglement significantly influences the mechanical response in sodium gels. Both modulus and tensile stress-strain responses exhibit noticeable variations with different degrees of entanglement.

[AA+SA]	M_0 (8.9M)	0.8 M_0 (7.1M)	0.67 M_0 (5.9M)	0.6 M_0 (5.3M)	0.5 M_0 (4.4M)
pH	2.43	2.42	2.48	2.52	2.56
Na ⁺					

Fig. R10 Different monomer concentrations were used to fabricate Na⁺-MPEC gels with different polymerization degree. The measured pH for each photoresin was included. Images of as-printed gels with the corresponding conditions.

Fig. R11 Quasi-static tensile test for the fabricated Na⁺-MPEC gels. All gels were post-processed to the same conditions described in the main text. **a** all samples, **b** zoomed-in axes for samples with lower concentration to show stress-strain curve shape change, and **c** summary of the measured young's modulus.

3 The coordinate bond strength is affected by the charged valence of the metal ion, but it also depends on metallic ions. For example, the Ca^{2+} , Zn^{2+} , and Fe^{2+} have different coordination bond strengths with carboxyl groups. Could you give some discussions on this point based on theoretical analysis? In addition, How about the case of tetravalent coordinating ions such as Zr(IV) and Ti(IV) , besides Na^+ , Ca^{2+} and Fe^{3+} ?

We thank the reviewer’s insightful point regarding the influence of metal characteristics. Indeed, the diverse properties of metallic ions, such as ionic radii and valency, are of great interest. Our quantum chemistry results in Figure 1d suggest significant differences in binding energy among various metal ions. To explore the distinct coordination strengths of metal ions in greater detail, we conducted molecular dynamics simulations with different metal ion radii and experimentally validated our findings. **These results are summarized in Section 2 of the revised SI.**

In brief for different metal identities with di- and trivalent metal species:

- We considered metal ionic radius and valency as representing different metal identities. From our quantum chemistry analysis, depicted in Figure 1d, binding energy scales with both metal ionic radii and valency.
- Our MD simulations revealed that stronger binding of smaller metal ions results in slower relaxation times for both ion-pairing and the end-to-end vector. This leads to a narrower cluster size distribution and increased stiffness under tensile loading.
- We experimentally explored these ion effects on mechanical properties by synthesizing gels with various metal ions, such as Ni, Zn for divalent, and Ga, Fe, and Cr for trivalent systems, following a similar trend with our theoretical predictions.
- It is noted that transition metals species with unfilled d-orbitals or complex geometries, need to consider more than binding strength as these additional factors may alter cluster formation and distribution. A more sophisticated computational model will be needed to accurately capture their intricate interactions.

While tetravalent ions present a fascinating avenue for exploration, we regretfully opted not to include discussions on them in this study due to several challenges encountered in both experimental and computational work:

- Firstly, obtaining the binding energy via quantum chemistry calculations for these high-valency ions proved highly non-trivial due to the complexity of their coordination environments, which sometimes involve multiple metal centers. Although simplified calculations involving a single acetate suggest stronger binding than trivalent ions, molecular dynamics simulations of tetravalent gels pose a challenge due to the intricate coordination environment.
- Additionally, relaxation timescales in Figure 3a indicate an order of magnitude longer times, necessitating computational resources beyond our current capabilities.

- In our experimental endeavors, we fabricated Zr-MPEC and conducted uniaxial testing. The results indicated superior performance of Zr-MPEC over Al-MPEC, suggesting that tetravalent ions induce stronger binding strength. However, we encountered challenges as the Zr resin solution precipitated over time due to the insolubility of Zr salts in the solvent. Given that resin inhomogeneity can significantly impact polymerization, metal density, and polymer network microstructure, we recognized the need for a more dedicated study to fully understand the contributions of tetravalent species.

Fig. R12 Stress-strain curves of MPEC gels with trivalent and tetravalent metal species, measured using quasi-static tensile tests.

We hope that the reviewer understands our decision, and we remain open to exploring this intriguing research avenue in future studies.

4 The structural mode of complexation of metal species is crucial, while the bidentate chelation relationship between each metal ion and the polymer has been elucidated through FTIR-ATR recordings referred to the Ref 20 and Ref 21. Are there any other characterization methods available to provide additional evidence for this conclusion?

Determination of the structural mode of complexation is frustrated in polymeric species owing to the sensitivity of either polymers to degradation with many standard techniques or owing to water loss in the gel which can change the structure irreversibly after its exotherm-endothrm transition (circa $\sim 150^\circ\text{C}$, see thermal characterization) from dehydration of the gel. The carboxylate-metal complexation mode determination by FTIR is well characterized technique, utilized in a range of fields from hydrometallurgy (Hydrometallurgy 72.1-2 (2004): 139-148), environmental science and surface chemistry (Environ. Sci. Technol. 1990, 24, 822-828), geochemistry (Chemical Geology 273.1-2 (2010): 55-75), biochemistry (Carbohydrate research 345.4 (2010): 469-473), molecular structure and spectroscopy (Spectrochimica Acta Part A: Molecular and Biomolecular Spectroscopy 140 (2015): 563-574; Journal of molecular structure 744 (2005): 705-709), and inorganic chemistry (Inorg. Chem. 2008, 47, 15, 6711-6725). Therefore, we remain confident that this analysis is sufficient for confirmation of coordination and relevant structural modes of complexation in an average sense. However, we understand the reviewer's desire for additional characterization methods to be sure of our conclusions made using FTIR. Therefore, we have pursued some alternative methods to provide some corroboration, though with noted caveats regarding their applicability below:

- Photospectroscopy – water is a weak field ligand and relatively stronger than carboxylate. Both compounds are pi-donors with small octahedral (O_h) splitting yielding high spin metal complexes. However, colorimetry in our gel system as non-destructive spectroscopic technique relies on presence of d-electrons to be present to see coordination. Shifts in the spectrum attributed to coordination by water vs. carboxylate or mixed systems can be deduced, coordination geometry can be ascertained by color. Ligand field splitting energy is expected to favor an O_h coordination which would favor bidentate chelation. All samples possessing color tested (eg. Cobalt II, Nickel II, Chromium III, Iron III) are in O_h geometries consistent with their colors in solution phase chemistry (Coordination Chemistry Reviews 135 (1994): 623-650), in metal compounds (color of metal compounds. CRC Press, 2000) and natural settings (eg. Mineralogical applications of crystal field theory, 1993).

Fig. R13 Solid-state UV-Visible photospectroscopy of relevant transition metal MPEC gels in transmission.

- Moessbauer Spectroscopy – While ^{57}Co -Moessbauer is only capable of measuring iron, it provides quantitative information regarding coordination, oxidation state, etc. relative to a native iron calibration standard. We have used this technique on our Fe(III) MPEC sample to measure an isomer shift of $\delta_{iso} \approx 0.528$ and a quadrupole splitting of $\Delta E_Q \approx 0.589$, consistent with *symmetric* octahedral coordination of the Fe^{3+} center (Zeit. Krist.-Cryst. Mat. 143.1-6 (1976):14-55; RCS 41 (2008):5603-5611; Nature 396.6712 (1998): 667-670). Further, examination of the resin (prior to photopolymerization) shows very similar values with $\delta_{iso} \approx 0.518$ and a quadrupole splitting of $\Delta E_Q \approx 0.544$. While this analysis is limited to Iron, in natural settings, trivalent coordination by Al^{3+} and other trivalent species is commonly substitutional with Fe^{3+} in natural aqueous (J. of Inor. Biochem., 44(2), 141-147), crystalline (Phys. B: Cond. Mat., 245(2), 119-122), and mineral systems (Geo. et Cosmo. Acta, 75(16), 4667-4683; Geo. et Cosmo. Acta, 220, 217-234). For this reason, we believe the coordination of Iron (III), which regularly undergo ionic substitutions in nature, is representative of our other trivalent systems. Therefore existing in a symmetric, octahedral coordination sphere implying bidentate chelation.

Fig. R14 ^{57}Co -Moessbauer spectra of Fe(III) gel phase and flash frozen resin (solution).

- X-Ray Absorption Spectroscopy (XAS) – X-Ray absorption spectra would be illuminating. The near edge spectrum (XANES) is sensitive to oxidation state and coordination chemistry (e.g., octahedral vs tetrahedral) of the absorbing atom, while the extended spectrum (EXAFS) can be used to determine bond distances and species neighboring the absorbing atom. While beamline experiments with our samples are beyond the scope of this work, we have performed preliminary XANES studies using EasyXAFS300+ system for bench top analysis. This instrument is limited in which elements are available, but we were able to run Zn^{2+} , Ni^{2+} , Fe^{3+} , and Co^{3+} among others. From these, XANES data shows good agreement with FFEF *ab initio* multi-scattering simulation results of di- and trivalent ionic states, respectively (Phys. Chem. Chem. Phys. (2010), 12, 5503-5513; Comptes Rendus Physique (2009), 10 (6) 548-559; Rev. Mod. Phys. (2000), 72, 621). Extended spectra collected on this instrument were insufficiently resolved to interpret in-detail, but the near edge spectrum is indicative of octahedral coordination environments in all MPEC gels the EasyXAFS instrument was capable of measuring.

A more rigorous treatment exceeds the scope of this work, however is of interest for future study using beam-line methods such as Pair Distribution Function combined with interpretative qDFT methods would be confirm identities of nearest neighbors, relative orientation and distances, and the secondary coordination shell. For now, however, initial XAS supports our FTIR conclusions regarding octahedral coordination which in a symmetric structure would again imply bidentate chelation of the metal center by the carboxylate.

Fig. R15 Near edge X-ray absorption (XANES) spectra of Zn^{2+} and Ni^{2+} plotted with reference metal films agreeing well with *FFEF* ab initio multi-scattering simulations.

- Solid-State Nuclear Magnetic Resonance Spectroscopy (ssNMR) – ssNMR would provide another alternative method for structural determination but was ruled out for our experimental methods in this review resonance owing to the extent of enrichment required for sufficiently high signal-to-noise ratio. With that said, we believe isotopic enrichment methods combined with 2D-ssNMR would prove useful for correlation of bonding environment from chemical shift as an alternative method of coordinating environment and determination of the complexes' geometry.

5 In Fig. 2b, the DMA and DSC results indicate a glassy-rubbery transition. If this is a gel, what is the water content? Why is this gel in a glassy state at relatively low temperatures? The Young's modulus in Fig. 2d of the gel is several MPa, much lower than the storage modulus in Fig. 2b. What's the reason?

As previously addressed in comment #1, the solvent content in our gels was determined through TGA analysis, revealing that all gels contain $\sim 14\% \pm 2.14\%$ water by weight. The inclusion of considerable amounts of solvent results in a much lower glass transition temperature for our system compared to the known T_g for the pure PAA-system.

Regarding the discrepancy between the Young's modulus reported in Fig. 2d and the storage modulus in Fig. 2b, it's important to note that in viscoelastic materials, there is no true Young's modulus representing the intrinsic elasticity of the material. Instead, we report the material response at the fixed loading rate (performed quasi-statically), capturing the elasticity at that particular time frame, and compare the modulus of different gels at the same loading rate for a fair comparison. In our study, Young's modulus is done quasi-statically, while the dynamic test is at higher frequency (1 Hz). Any differences in applied loading rates between the measurements can result in discrepancies between the modulus values. The larger storage modulus than the observed young's modulus suggests that our gels are highly viscoelastic, with the applied time frame of the quasi-static tensile test being slower than that of the dynamic testing.

6 It is very meaningful that MD results can distinct the cluster size of the gels with different metal ions. What's the reason for the formation of cluster, and which factor determines the size? What's the relation of the strength between clusters and single coordinate bond?

We appreciate the insightful inquiry from the reviewer. In our molecular dynamics simulations, clusters emerge due to favorable enthalpic interactions between the charged beads on the polymer backbone and the multivalent metal ions, resulting in what we term dynamic crosslinks. These clusters predominantly form through inter-crosslinking between charged sites on different chains rather than intra-crosslinking within the same chain. The balance between intra- and inter-crosslinks, governed by entropic considerations, determines the cluster formation and ultimately influences their sizes. For instance, as demonstrated in Figure 3c, both the metal ion valency and the polyanion charge density impact the distribution of inter- versus intra-crosslinks in the system. While higher-valency metal ions have a greater propensity to form dynamic crosslinks, this can be balanced out by a lower charge sparsity (or higher pH-gels), which favors intra-crosslinks and does not contribute to cluster formation.

In terms of the strength of a single coordinate bond for the same valency, we have shown in Section 2 of the revised SI, that, by increasing the bond strength (or reducing the metal ion size), for the same charge sparsity, there is indeed an increase in the longevity of crosslinks which, unsurprisingly, leads to larger, longer-lasting clusters. Thus, the increased likelihood of forming dynamic crosslinks in general increases the likelihood of forming inter-crosslinks as well. This relationship is not perfectly linear as there will be a point where all the metal ions will be cross-linking, with no further increase when increasing the binding strength.

7 Fig. 4d shows interesting bending behavior of the gel at different strain rate. This result is explained in terms of viscoelasticity of the gel with dense physical associations. It should be better to study the rheological behavior and correlate with the mechanical performance of the gel.

The reviewer raises a relevant point regarding the study of rheological behavior in our gels. We have conducted experiments using dynamic mechanical analysis (DMA) to investigate the modulus dependency on the applied frequency at fixed temperature, revealing significant time-dependent behavior for both Ca and Al-gels. However, upon examining the storage and loss moduli, we found that obtaining the full spectrum of time scales required to probe polymer relaxation times was not attainable within our experimental setup. Instead, to capture the time-dependent properties of the system, we included stress relaxation and strain recovery experiments, as shown in Supplementary Fig. 12. These experiments demonstrate different relaxation timescales with different metal ions or charge sparsity, aligning well with our observations in Figure 4d.

Fig. R16 Measured storage modulus and loss factor of Ca^{2+} and Al^{3+} -MPEC using rheological frequency sweep at room temperature from DMA. The data indicate that both gels are highly viscoelastic and Ca^{2+} is more rate-dependent than Al^{3+} .

Reviewer 3

Number	Comment	Response
0	The authors prepare an aqueous solution of monomers, initiator, and ions. They vary pH and concentration of ions of different valencies (Na¹⁺, Ca²⁺, and Al³⁺) in the precursor. The authors then use Liquid Crystal Display stereolithography, through radical polymerization, to fabricate hydrogels. These hydrogels are crosslinked through ionic interactions. They characterize the hydrogels in many ways (stress-strain curves, toughness, Fourier transform infrared spectroscopy, X-ray fluorescence Microscopy, differential scanning calorimetry, optical microscopy, thermal gravimetric analysis, dynamic mechanical analysis, scanning electron microscopy, and confocal microscopy). They conduct numerical simulations to predict the strength of ionic interactions of crosslink and inter-chain/intrachain crosslink ratio of network topology. The authors also construct phase diagrams to help identify the required synthesis conditions to fabricate tough hydrogels. The combination of experiments of many varieties, as well as calculations, makes this work unusually comprehensive among works on the mechanical behavior of hydrogels. The experiments and calculations shed insight on ionically crosslinked polymer hydrogels. Furthermore, the method of fabrication enables the production of structures of complex shapes. The authors demonstrate the gel recyclability, mechanical response of bilayer structure, and the morphing of the structures in response to temperature and solvent.	We sincerely appreciate the reviewer's thoughtful summary of our work and their recognition of its potential significance.

1 The ionically crosslinked polymer hydrogels have been studied extensively. In the abstract, the authors state “We discover that the mono-, di-, and trivalent metal ions afford control of the coordination environment and bond strength within the polymer matrix, which propagates to the macroscale properties where higher valency ions result in stiffer and tougher materials.” It is unclear which part is a discovery. The effect of multivalent ions on mechanical properties of polyelectrolytes have been studied in an enormous number of papers. Some examples are listed below. In light of these papers and many other papers in the literature on the ionically crosslinked polymer hydrogels, the authors need to review this literature and make a clear statement of the novelty of the paper.
Redacted for clarity

We deeply appreciate the reviewer’s insightful observation. Upon careful consideration and a thorough examination of the literature, we acknowledge the need for a clearer articulation of the novelty of our work within the abstract. While it is indeed well-documented that multivalent ions can influence the mechanical properties of polyelectrolytes, we recognize that our contribution lies in the development of a comprehensive framework, informed by both theoretical insights and experimental data, to systematically design and tailor the material properties of our mono-, di-, and trivalent metal-polyelectrolyte complex gels.

To better convey this unique aspect of our research, we have refined the abstract including the relevant statement to underscore our contributions effectively. Our revised abstract is (The modified sentences were highlighted in red):

Metallo-polyelectrolytes have emerged as compelling functional materials for various applications — from filtration to biomedical devices and sensors — through the unique metal-organic matrix synergy [1 – 4]. The dynamic, reversible electrostatic interactions between metal ions and charged polymer chains offer advanced functionality such as relatively high ionic conductivity, thermal and electrochemical stability, self-healing capabilities, and broad tunability of mechanical properties in this unique class of materials [5]. However, the knowledge gap between molecular-level chemistry of dynamic bonds and continuum-level material properties persist, largely due to limited fabrication methods and a lack of theoretical design frameworks, hindering the development and utilization of these materials in real-world applications [5]. Existing state-of-the-art fabrication methods typically involve cumbersome, multi-step solution-based synthesis, which usually produces materials with inhomogeneity and poor long-term stability. The multitude of time/length scales associated with the actuation and stimulus-driven response of metallo-electrolytes presents further computational challenges, which limits theoretical guidance for the development of experimental methods. To address this critical gap, we present a comprehensive framework informed by both theoretical insights and experimental findings, elucidating the remarkable interplay of molecular parameters in governing material properties of MPEC gels. Leveraging a single-step stereolithography-based additive manufacturing (AM) method, we demonstrate the production of homogeneous, stable, and long-lasting MPEC gels, providing a straightforward synthesis route. These AM-fabricated MPEC gels offer facile tunability in their mechanical response via two critical control parameters: (1) metal ion valency and (2) polymer charge sparsity. Using these control parameters, we investigate how the mono-, di-, and trivalent metal ions afford control of the coordination environment and bond strength within the polymer matrix.

We also analyze how polyanion charge sparsity, regulated by the pH of the precursor photoresin, impacts the gel phase. Through an experimentally-informed, theoretically-guided approach, we unveil mechanistic insights into how these molecular interactions propagates to the macroscale properties where higher valency ions result in stiffer and tougher materials while lower charge sparsity gels lead to changes in the material phase behavior. This work provides a comprehensive understanding of the metallo-polyelectrolyte behavior, laying out their parameter space and enabling selective design of advanced compliant and functional metallo-polyelectrolytes.

Additionally, we have revised the introduction of our paper to provide a more thorough context for our work, integrating references provided by the reviewer.

We hope that this reframing of the story better conveys the value and the novelty of our study. Through this review process, we have deepened our understanding of the design parameters governing MPEC gels, thereby facilitating their customization to meet diverse application needs.

Reviewers' Comments:

Reviewer #1:

Remarks to the Author:

Overall, the authors have made a strong effort to address the concerns raised by me and other reviewers. The experimental section of the manuscript is improved. I am supportive of publication.

Reviewer #2:

Remarks to the Author:

My concerns have been addressed. The authors made careful revisions to the manuscript that improved the overall quality. I think this paper deserves publication in Nat. Commun. Two small issues should be considered.

1. The length and format of the Abstract should fit the requirement of this journal. The current abstract seems too long. The refs should be not included in the abstract.

2. The gels have a relatively low water content, i.e., about 15 wt.%. Is it appropriate to term these materials as gels? There are many hydrogels physically crosslinked by metal-ligand coordinates, having much higher water content. Why the gels in this work has such a low water content?

Response to Reviewers

We would like to express our gratitude to the reviewers for their insightful feedback and contributions. We are delighted that both found the manuscript interesting. Below, we have addressed all of the reviewers' comments and have highlighted any changes to the manuscript in red within the main body.

Reviewer 1		
Number	Comment	Response
0	Overall, the authors have made a strong effort to address the concerns raised by me and other reviewers. The experimental section of the manuscript is improved. I am supportive of publication.	We are delighted to hear that the reviewer is pleased with our improvements to the manuscript and is supportive of its publication. We greatly appreciate the reviewer's valuable feedback and contributions, which have been instrumental in enhancing the quality of our work.

Reviewer 2		
Number	Comment	Response
0	My concerns have been addressed. The authors made careful revisions to the manuscript that improved the overall quality. I think this paper deserves publication in Nat. Commun. Two small issues should be considered.	We appreciate that the reviewer agrees our manuscript deserves publication. We appreciate the positive feedback and their valuable contributions to improving our work. In response to their suggestions, we have addressed the comments below.
1	The length and format of the Abstract should fit the requirement of this journal. The current abstract seems too long. The refs should be not included in the abstract.	We are grateful to the reviewer for bring up an important point. We have removed the references and revised it to adhere to the journal's word limit requirements. Our revised abstract (147 words) is:

Metallo-polyelectrolytes are versatile materials for applications like filtration, biomedical devices, and sensors, due to their metal-organic synergy. Their dynamic, reversible electrostatic interactions offer high ionic conductivity, self-healing, and tunable mechanical properties. However, the knowledge gap between molecular-level dynamic bonds and continuum-level material properties persists, largely due to limited fabrication methods and a lack of theoretical design frameworks. To address this critical gap, we present a framework, combining theoretical and experimental insights, highlighting the interplay of molecular parameters in governing material properties. Using stereolithography-based additive manufacturing, we produce durable metallo-polyelectrolytes gels with tunable mechanical properties based on metal ion valency and polymer charge sparsity. Our approach unveils mechanistic insights into how these interactions propagate to macroscale properties, where higher valency ions yield stiffer, tougher materials, and lower charge sparsity alters material phase behavior. This work enhances understanding of metallo-polyelectrolyte behavior, providing a foundation for designing advanced functional materials.

2 The gels have a relatively low water content, i.e., about 15 wt.%. Is it appropriate to term these materials as gels? There are many hydrogels physically crosslinked by metal-ligand coordinates, having much higher water content. Why the gels in this work has such a low water content?

We appreciate the reviewer for the critical observations. It is correct that our gels have a relatively low water content (~ 15 wt.%). We intentionally avoid classifying this material as a hydrogel due to the small amount of water present. Instead, in our study, we define gels based on the presence of a percolating polymer network, supported by molecular dynamics simulations and confocal imaging. The inclusion of glycerol (~ 10 vol.%) serves as an additional plasticizer to facilitate polymer and ion mobility within the material. The lower water content is primarily due to the high monomer concentration used in our fabrication process, the specific printing technique employed, and the equilibration process which allows for some water evaporation over several days. These factors collectively contribute to the unique properties and structural characteristics observed in our gels.